# Subnanometric alkaline-earth oxide clusters for sustainable nitrate to ammonia photosynthesis

Jieyuan Li[1,2], Ruimin Chen[1], Jielin Wang[1], Ying Zhou [3], Guidong Yang[4] & Fan Dong [1,2 ✉]

The limitation of inert $N_2$ molecules with their high dissociation energy has ignited research interests in probing other nitrogen-containing species for ammonia synthesis. Nitrate ions, as an alternative feedstock with high solubility and proton affinity, can be facilely dissociated for sustainable ammonia production. Here we report a nitrate to ammonia photosynthesis route ($NO_3^-$RR) catalyzed by subnanometric alkaline-earth oxide clusters. The catalyst exhibits a high ammonia photosynthesis rate of 11.97 mol $g_{metal}^{-1}$ $h^{-1}$ (89.79 mmol $g_{cat}^{-1}$ $h^{-1}$) with nearly 100% selectivity. A total ammonia yield of 0.78 mmol within 72 h is achieved, which exhibits a significant advantage in the area of photocatalytic $NO_3^-$RR. The investigation of the molecular-level reaction mechanism reveals that the unique active interface between the subnanometric clusters and $TiO_2$ substrate is beneficial for the nitrate activation and dissociation, contributing to efficient and selective nitrate reduction for ammonia production with low energy input. The practical application of $NO_3^-$RR route in simulated wastewater is developed, which demonstrates great potential for its industrial application. These findings are of general knowledge for the functional development of clusters-based catalysts and could open up a path in the exploitation of advanced ammonia synthesis routes with low energy consumption and carbon emission.

[1] Research Center for Environmental and Energy Catalysis, Institute of Fundamental and Frontier Sciences, University of Electronic Science and Technology of China, Chengdu 611731, China. [2] Yangtze Delta Region Institute (Huzhou), University of Electronic Science and Technology of China, Huzhou 313000, China. [3] School of New Energy and Materials, Southwest Petroleum University, Chengdu 610500, China. [4] XJTU-Oxford Joint International Research Laboratory of Catalysis, School of Chemical Engineering and Technology, Xi'an Jiaotong University, Xi'an 7010049, China. ✉email: dongfan@uestc.edu.cn

As an essential chemical, ammonia ($NH_3$) is industrially produced via the Haber-Bosch process, which consumes 1.0–2.0% of the world's energy output and contributes to 1.6% of the world's carbon emissions[1–4]. As an alternative, artificial electro-/photo-/photoelectrochemical nitrogen reduction reactions ($N_2$RRs) for $NH_3$ synthesis, inspired by the natural microbial $N_2$ fixation, have attracted tremendous research interest[5,6]. Despite great achievements in recent decades, it is inconvenient to overlook that the future of $N_2$RRs is plagued by the ultrahigh dissociation energy of the N≡N bond (941 kJ $mol^{-1}$)[7,8]. Inferior catalytic performance is predictable, arising from the limited solubility and low proton affinity of the inert $N_2$.

From an energy viewpoint, nitrate ions ($NO_3^-$), as a sustainable N-containing alternative, can be disintegrated at lower dissociation energy of 204 kJ $mol^{-1}$, contributing to an accelerated reaction kinetics for $NH_3$ synthesis[9–13]. Besides, the highest valence state of N-element in $NO_3^-$ ensures that the deep reduction reaction can be achieved for selective $NH_4^+$ synthesis. The intermediate-valence $N_2$ oxidation and reduction may proceed simultaneously when conducting $N_2$RR, which restrains the $NH_4^+$ selectivity[14–16]. Another virtue of using $NO_3^-$ feedstock lies in its rich distribution in wastewater. The abundant nitrate in wastewater offers sufficient reactants for $NO_3^-$ reduction reaction ($NO_3^-$RR) routes[17–20]. Instead of the partial reduction of $NO_3^-$ to $N_2$ for its purification, the eight-electron transfer reaction for $NO_3^-$ to $NH_4^+$ synthesis provides an opportunity for the value-added conversion of contaminative $NO_3^-$ into ammonia as an economically competitive product. Also, the wide distribution of general organic matters such as aldehydes and phenols in wastewater is noted, which forms contaminant mixtures with nitrate[21]. These organic matters can serve as the hole sacrificial agents, which accelerates both the $NO_3^-$ reduction for $NH_4^+$ synthesis and pollutants' oxidation for their degradation. Thus, the development of a $NO_3^-$RR route for $NH_4^+$ production, which provides sustainable N-cycle utilization, has a profound effect in both reducing energy consumption and mitigating environmental anxieties.

Despite its advantages, $NO_3^-$RR also suffers from some inevitable difficulties as an active, yet challenging area of current research. In the eight-electron transfer reaction for $NO_3^-$-$NH_4^+$ synthesis, competitive side reactions may be fierce, mainly ascribed to five-electron transfer for the partial reduction of $NO_3^-$ to $N_2$ and hydrogen evolution reaction (HER)[22–24]. Moreover, it is well established that the yield rate and selectivity for $NH_4^+$ dominantly rely on the development of novel catalytic materials, precise regulation of the reaction parameters and systematic investigation of the reaction mechanism. In this scenario, a comprehensive catalysis system requires a rational design to sufficiently promote catalytic performance.

As a typical solid–liquid heterogeneous catalytic reaction, the interaction between catalysts and solvents is essential in the $NO_3^-$RR system. Generally, metal cations ($M^{x+}$) are introduced into solvents, serving as key functional components such as cocatalysts, ionic liquids or electrolytes[25–27]. With the catalytic reaction on stream, the dynamic evolution of $M^{x+}$ can be observed[28,29]. With the assistance of appropriate catalyst support and reaction conditions, the deposition of $M^{x+}$ on a solid surface is expected, which leads to the production of corresponding single atoms, nanoclusters or nanoparticles on a substrate surface[30–32], thereby modifying the interfacial structure and in situ serving as the catalytic active sites. Key challenge lies in revealing the interfacial structure of active sites for facilitating the ammonia selectivity and suppressing the occurrence of side reactions (HER and $NO_3^-$ to $N_2$) to enhance the efficiency.

Here, we demonstrate a general strategy to accomplish the *operando* construction of subnanometric alkaline-earth oxide clusters ($MO_{NCs}$, M=Mg, Ca, Sr or Ba) as the active sites due to the nontoxicity and low price of alkaline-earth metals. Also, the widely investigated $TiO_2$ nanosheets (TNS) is applied as the substrate since it is easy to be fabricated and characterized. After the *operando* construction of the $BaO_{NCs}$-TNS composite, a high ammonia photosynthesis rate of 11.97 mol $g_{metal}^{-1}$ $h^{-1}$ (89.79 mmol $g_{cat}^{-1}$ $h^{-1}$) is reached with nearly 100% selectivity. A total ammonia yield of 0.78 mmol within 72 h is achieved. The local interfacial structure is precisely tailored to strengthen charge transfer at the $MO_{NCs}$/TNS interface. Then, it is revealed that the eight-electron transfer reaction for $NO_3^-$RR is notably accelerated to achieve a high rate for sustainable $NH_4^+$ photosynthesis. The practical application of $NO_3^-$RR route in simulated wastewater is also developed, which establishes an intriguing "sacrificial-agent-free" route for ammonia synthesis and demonstrates great potential for its real industrial application. The current ammonia photosynthesis route could provide an alternative route for nitrogen cycle utilization and promote the development of low-carbon technology.

## Results

**Operando construction of the subnanometric clusters.** Alkaline-earth ions (50 mg/L) are first injected into the reaction mixture for the $NO_3^-$RR on TNS. Scanning electron microscopy (SEM), transmission electron microscopy (TEM) and X-ray diffraction (XRD) results demonstrate that the morphology and crystal structure of TNS is well maintained after alkaline-earth ion incorporation (Supplementary Figs. 1–3). To reveal the *operando* evolution of the alkaline-earth ions, the mixture is extracted from the reaction on stream. As identified by the quasi in situ high-angle annular dark-field scanning transmission electron microscopy (HAADF-STEM), the *operando* evolution of the alkaline-earth species on the catalyst surface is observed (Fig. 1).

Taking $Ba^{2+}$ as an example, once the $BaCl_2 \cdot 2H_2O$ is introduced into the reaction mixture, single atoms ($Ba_{SAs}$) are constructed on the TNS surface after 5 min of irradiation (Fig. 1a). The subsequent growth and agglomeration of $Ba_{SAs}$ lead to the generation of subnanometric BaO clusters ($BaO_{NCs}$) with a size of ~0.6 nm (Fig. 1b). With prolonged irradiation time (Fig. 1c, d for 60 and 120 min respectively), uniformly dispersed $BaO_{NCs}$ with a mean size of 0.7 ± 0.3 nm are formed on TNS. In addition, the HAADF-STEM elemental mappings (Fig. 1e) confirm that the $MO_{NCs}$ are mainly composed of Ba. It is worth noting that the variation of the $Ba^{2+}$ concentration is revealed by ion chromatography (IC) detection (Fig. 1f). A continuous decrease in the $Ba^{2+}$ concentration is observed within the first 60 min of irradiation. Then, equilibrium is reached to guarantee the subnanometric size of the $BaO_{NCs}$, thereby preventing excessive agglomeration and further growth. In addition, the *operando* construction of $MgO_{NCs}$, $CaO_{NCs}$ and $SrO_{NCs}$ is accomplished under the same procedure as that for the $BaO_{NCs}$ (Supplementary Figs. 4–8), indicating that this is a general strategy to form the $MO_{NCs}$.

**Characterization and electronic properties of the $MO_{NCs}$-TNS composites.** The chemical components and valence states of the $BaO_{NCs}$ were investigated by X-ray photoelectron spectroscopy (XPS, Fig. 2a and Supplementary Fig. 9). The deconvolution of the Ba 3d XPS spectrum illustrates that the four characteristic peaks are fitted at the binding energies of 794.61, 792.84, 778.99, and 777.15 eV. The peaks located at 792.84 and 777.15 eV correspond to the spin orbits of Ba $3d_{3/2}$ and Ba $3d_{5/2}$ respectively, demonstrating the generation of $BaO_{NCs}$ on the TNS surface. The other two shoulder peaks (794.61 and 778.99 eV) are identified as

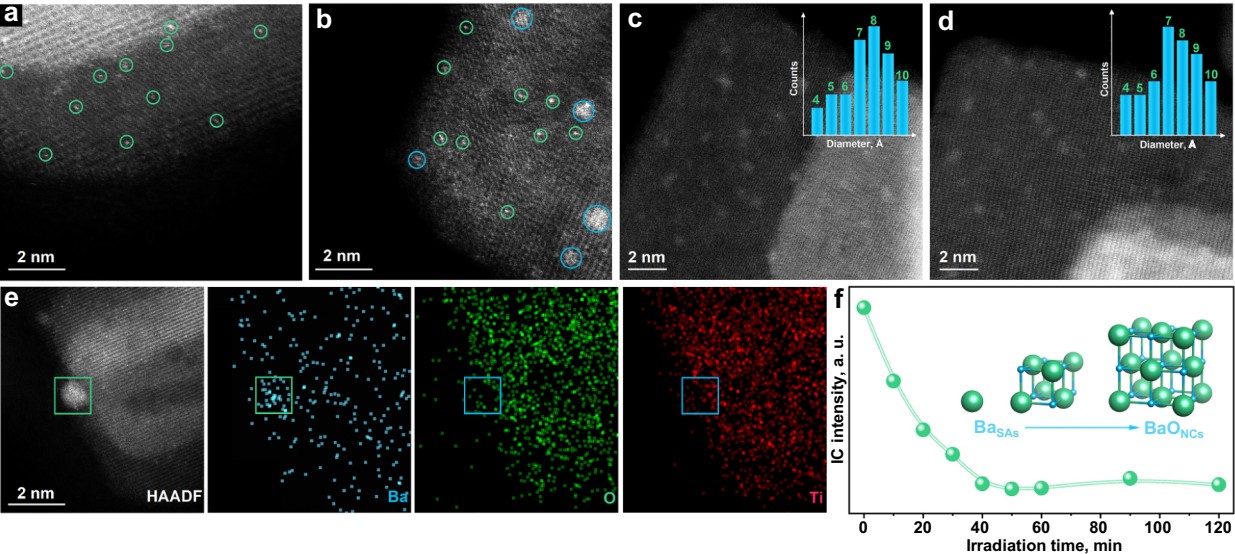

**Fig. 1 Structure identification of subnanometric BaO nanoclusters (BaO$_{NCs}$) *operando* construction on TiO$_2$ nanosheets (TNS) support. a–d** Quasi in situ high-angle annular dark-field scanning transmission electron microscopy (HAADF-STEM) images showing the evolution course from isolated Ba single atoms (Ba$_{SAs}$) to subnanometric (BaO$_{NCs}$) at the irradiation time of 5 min (**a**), 10 min (**b**), 60 min (**c**) and 120 min (**d**) respectively. The related size distribution is labeled as insets (**c**, **d**), in which the range of both *x* (0.4–1.0) and *y* (0–7) axes are set consistently. **e**, HAADF-STEM image (left) and respective elemental mapping images (right) verifying the component of Ba elements on the BaO$_{NCs}$-TNS surface. **f** Variation of Ba$^{2+}$ concentration during the *operando* construction of BaO$_{NCs}$ detected by ion chromatography.

Ba–O bond formation between the Ba in the BaO$_{NCs}$ and the O in the TNS[33,34]. The concentration of Ba is determined to be 0.75 wt.% and 0.26 at.% by using X-ray fluorescence (XRF) spectroscopy (Supplementary Fig. 10). To investigate the underlying growth pattern and mechanism of BaO$_{NCs}$, electron paramagnetic resonance (EPR) measurements were conducted for the pristine TNS before and after light irradiation (Fig. 2b and Supplementary Fig. 11). The intensified signal for lone-pair electrons after light irradiation discloses that the oxygen vacancies in TNS can be constructed via light irradiation, which agrees with the reported results[35–37]. Furthermore, it is confirmed by density function theory (DFT, Fig. 2c) calculations that the construction of BaO$_{NCs}$ at the defective site of TNS is more energy-favorable (−0.69 eV) than that of pristine TNS (−0.36 V). A uniform pattern for the deposition of other alkaline-earth MO$_{NCs}$ on TNS is confirmed (Supplementary Figs. 12–15), which indicates that the subnanometric MO$_{NCs}$ can be precisely immobilized at the light-induced vacancy sites on TNS with this general method. Since the number of lone-pair electrons is limited at the vacancy sites, the cluster size is restricted at the subnanometric region, which hampers their excessive agglomeration and growth, thus achieving *operando* construction of subnanometric MO$_{NCs}$ at the defective sites of TNS.

Next, the optical and electronic properties of the as-prepared subnanometric MO$_{NCs}$-TNS composites were surveyed. Steady and time-resolved photoluminescence spectra (Fig. 2d and Supplementary Figs. 16 and 17) show that the charge separation capacity of TNS is observably enhanced after the *operando* construction of MO$_{NCs}$. The prolonged carriers' lifetime is beneficial for the participation of photoinduced electrons to catalyze the NO$_3^-$-NH$_4^+$ photosynthesis reaction. Besides, the red shift of the light absorption range (Fig. 2d inset and Supplementary Fig. 18) implies that the light capture and utilization ability is facilitated via the deposition of MO$_{NCs}$. Then the Mott-Schottky spectra and UV–vis diffuse reflection spectra (DRS) were combined to determine the band structures of TNS and BaO$_{NCs}$-TNS (Supplementary Fig. 19). It is noted that the conduction band position of BaO$_{NCs}$-TNS is elevated than that of

the pristine TNS, which could enhance the reduction capacity for the elevated NO$_3^-$-RR performance. Molecular-level insights into the charge transfer patterns at MO$_{NCs}$/TNS interface were further revealed by DFT calculations (Fig. 2e and Supplementary Figs. 20–22). As supported by both standard Perdew-Burke-Ernzerhof (PBE) functional and PBE + *U* correction calculations for the planar average potential energy profile, a distinct energy gap is generated between the BaO$_{NCs}$ and TNS surface, which facilitates directional electron migration from the BaO$_{NCs}$ to TNS. It is also confirmed by the charge difference distribution (Fig. 2e inset) that intense charge clouds accumulate at the BaO$_{NCs}$/TNS interface, building a unique electronic channel to promote charge transfer. Due to the superior photochemical properties contributed by the *operando* construction of the MO$_{NCs}$-TNS composite, the elevated catalytic performance of the NO$_3^-$-RR for NH$_4^+$ photosynthesis is expected.

**NO$_3^-$ to NH$_4^+$ photosynthesis performance.** The evaluation of the NO$_3^-$-RR for NH$_4^+$ photosynthesis was first conducted in 100 mL of KNO$_3$ solution (20 mg/L of NO$_3^-$) containing 5.0% ethylene glycol (EG) as the hole sacrificial agent under full-spectrum illumination. Briefly, 5.0 mg of pristine TNS is applied as the catalyst support, in which 50.0 mg/L alkaline-earth ions are injected. As depicted in Fig. 3a, the pristine TNS exhibits nice catalytic activity (1.65 mmol g$_{cat}^{-1}$ h$^{-1}$). The oxygen vacancies (OVs) in TNS are identified as the active sites due to the observable OVs construction via light irradiation (Fig. 2b and Supplementary Fig. 11). Most importantly, the *operando* construction of MO$_{NCs}$ (illustrated in Fig. 1f) and enhancement of the NH$_4^+$ photosynthesis rate are simultaneously accomplished with the reaction on stream. Since the construction of MO$_{NCs}$ and NH$_4^+$ synthesis proceed at the same time, the NH$_4^+$ synthesis rate by MO$_{NCs}$ is elevated and the gradual increase of the slope for NH$_4^+$ generation is reasonable. The NH$_4^+$ synthesis rate is increased from 1.65 mmol g$_{cat}^{-1}$ h$^{-1}$ with pristine TNS to 3.78 mmol g$_{cat}^{-1}$ h$^{-1}$ with BaO$_{NCs}$-TNS, which demonstrates the significant advantage of subnanometric MO$_{NCs}$ as cocatalyst. The apparent quantum efficiency (Supplementary Note 1) for these as-prepared

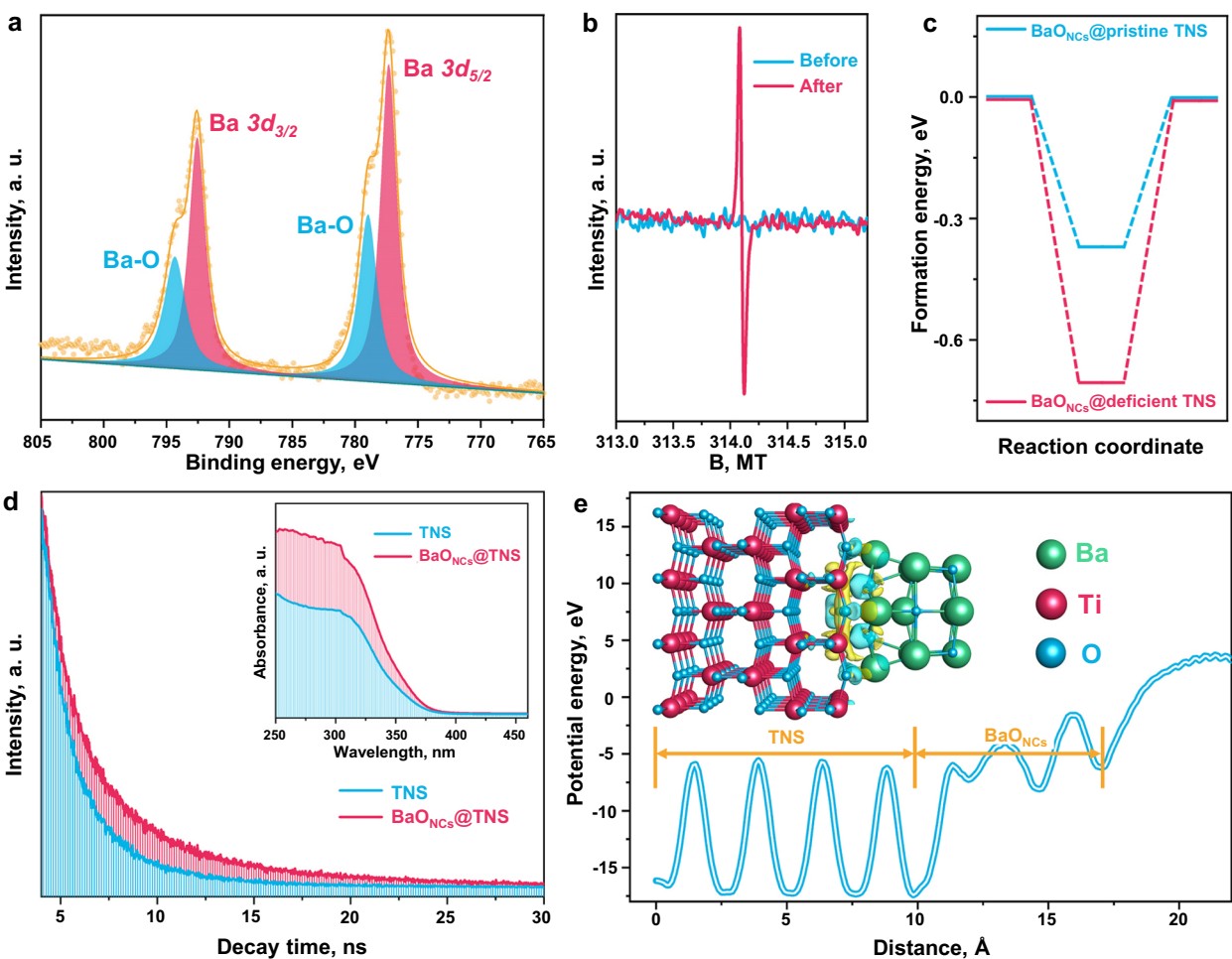

**Fig. 2 Chemical composition and electronic structure. a** Ba *3d* X-ray photoelectron spectroscopy (XPS) spectra of BaO$_{NCs}$-TNS. **b** Room temperature solid electron paramagnetic resonance (EPR) results of TNS before and after irradiation for 30 min. **c** Calculated binding energy of BaO$_{NCs}$ deposited at pristine and deficient TNS surfaces respectively. **d** Time-resolved fluorescence emission decay spectra. Inset: UV–vis diffuse reflection spectra (DRS) results. **e** Calculated planer average potential energy profile using Perdew-Burke-Ernzerhof (PBE) + *U* correction. Inset: calculated charge difference distribution at the BaO$_{NCs}$/TNS interface, in which charge accumulation is marked in blue and charge depletion is marked in yellow. The isosurfaces were set to 0.005 eV Å$^{-3}$.

samples is also enhanced in the order of TNS (1.86%) < CaO$_{NCs}$-TNS (2.59%) < MgO$_{NCs}$-TNS (3.09%) < SrO$_{NCs}$-TNS (3.22%) < BaO$_{NCs}$-TNS (3.46%). In addition, the controlled experiment is conducted by adding KCl into the catalysis system of pristine TNS without other cations or anions (Supplementary Fig. 23), which excludes the potential involvement of Cl$^-$ from the source of alkaline earth source (MCl$_2$·xH$_2$O). It is observed that the NO$_3^-$RR to ammonia efficiency is not promoted by the addition of Cl$^-$, which identifies that the enhanced activity is contributed by the construction of MO$_{NCs}$-TNS. Since the *operando* production of MO$_{NCs}$ on TNS is preferable to achieve at the OVs sites (Fig. 2c and Supplementary Figs. 12–15), the active sites in MO$_{NCs}$-TNS are regarded as the MO$_{NCs}$@OVs interfaces. Besides, to further unveil the activity origin of MO$_{NCs}$-TNS, we conduct an additional control experiment by replacing the TNS substrate with inert SiO$_2$ nanoparticles (Supplementary Fig. 24). It is observed that no NH$_4^+$ can be detected during the simultaneous construction of MO$_{NCs}$-SiO$_2$ and NO$_3^-$RR. Hence, it is clarified that the enhanced NH$_4^+$ synthesis efficiency gives credit to the construction of MO$_{NCs}$-TNS composites.

Because the catalytic efficiency is directly related to reaction parameters, the reaction parameters are comprehensively optimized to further promote catalytic performance (Fig. 3b),

including the NO$_3^-$ concentration, catalyst dose, and light source. It is observed that the increase in NO$_3^-$ concentration (100.0 mg/L) is beneficial for accelerating the reaction kinetics of NO$_3^-$RR (Supplementary Fig. 25). The saddle point of the catalyst dose is located at 1.0 mg to accomplish the optimal unit activity (Supplementary Fig. 26). Besides, since NO$_3^-$ could be preactivated by UV light, which tailors the coordination environment of the stable NO$_3^-$ and drives it into some active intermediates such as monodentate NO$_3^-$, –NO$_2$ and NO$_2^-$ (Supplementary Fig. 27), the utilization of different light sources was tested. Notably, the optimal NH$_4^+$ photosynthesis rate is 38.00 mmol g$_{cat}^{-1}$ h$^{-1}$ with pristine TNS after regulating the reaction parameters (Supplementary Fig. 28). Moreover, an accelerated rate for NH$_4^+$ photosynthesis is accomplished on BaO$_{NCs}$-TNS (89.79 mmol g$_{cat}^{-1}$ h$^{-1}$) with the reaction parameters of 100 mg L$^{-1}$ of NO$_3^-$, 1.0 mg of BaO$_{NCs}$-TNS catalyst, and UV light irradiation. It is worth mentioning that the introduction of UV light not only increases the energy density but also realizes the preactivation of NO$_3^-$, which exceeds that of the full-spectrum (15.80 mmol g$_{cat}^{-1}$ h$^{-1}$) and simulated solar light (3.07 mmol g$_{cat}^{-1}$ h$^{-1}$). Then, as shown in the XRF results (Supplementary Fig. 10), 0.75 wt.% of Ba element is detected in the BaO$_{NCs}$-TNS composite. Hence, the rate (per Ba metal) is

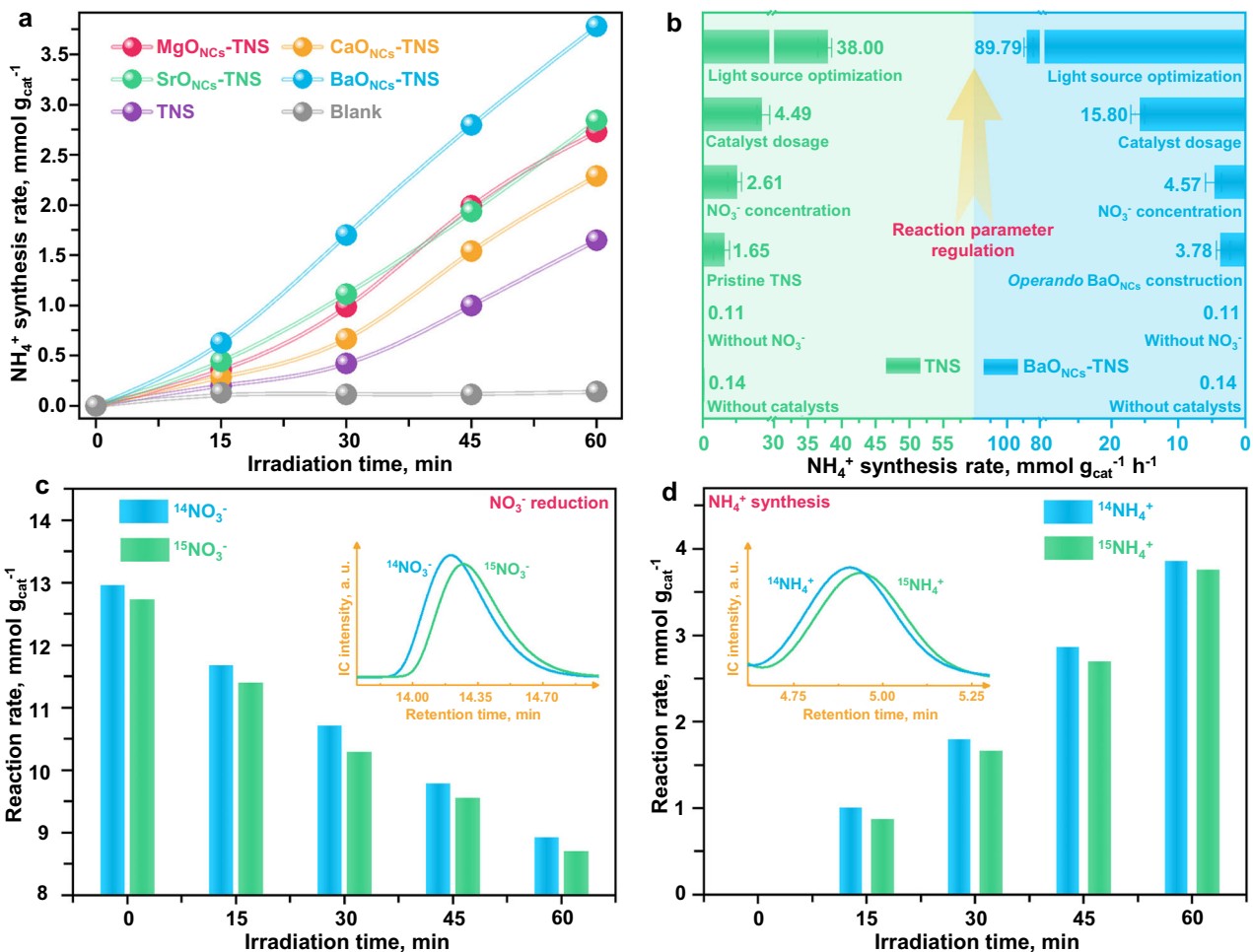

**Fig. 3 Catalytic performance of NH₃ photosynthesis on alkaline-earth oxide clusters. a** Catalytic efficiency tests showing the enhancement of *operando* construction of alkaline-earth oxide clusters on TNS surfaces. **b** Reaction parameter regulation for optimized NH₄⁺ photosynthesis rates on TNS and BaO_NCs-TNS respectively. **c**, **d** Quantitative isotope-labeled ¹⁵NO₃⁻ study verifying the fed NO₃⁻ as the source for the produced NH₃. Inset: raw ion chromatography (IC) spectra for ¹⁴NO₃⁻/¹⁵NO₃⁻ reduction (**c**) and ¹⁴NH₄⁺/¹⁵NH₄⁺ generation (**d**) respectively. The y-axis of **c**, **d** depict the reaction rates for NO₃⁻ reduction and NH₄⁺ production respectively. The inset images are the raw IC data for ¹⁴NO₃⁻/¹⁵NO₃⁻ (**c**) and ¹⁴NH₄⁺/¹⁵NH₄⁺ (**d**) respectively. The error bar was drawn based on the calculated standard error of two parallel tests.

calculated to be 11.97 mol $g_{Ba}^{-1}$ h$^{-1}$. The optimal rate catalyzed by BaO_NCs-TNS manifests advances in comparison with that of the other ammonia synthesis by using alkaline-earth-containing catalysts (Supplementary Table 1).

To exclude the potential contribution of contaminative N-containing species, blank control experiments were subsequently performed (Supplementary Fig. 29) under the same testing procedure as that without catalysts and NO₃⁻; these experiments confirm that almost no NH₄⁺ is produced. Most importantly, a quantitative isotope measurement was executed to test the N source for generating NH₄⁺ by combining IC and nuclear magnetic resonance (NMR) technologies (Fig. 3c, d and Supplementary Figs. 30–32). K¹⁴NO₃ and K¹⁵NO₃ solutions are used as N sources. Within 60 min of reaction, ¹⁵NH₄⁺ is observably detected when ¹⁵NO₃⁻ is employed. In addition, comparable rates for ¹⁴NO₃⁻/¹⁵NO₃⁻ reduction (Fig. 3c) and ¹⁴NH₄⁺/¹⁵NH₄⁺ production (Fig. 3d) are identified, confirming that the produced ammonia is directly derived from the nitrate feedstock, thus, the contribution of contaminative N-containing species to ammonia can be neglected.

We further investigated the selectivity of the NO₃⁻RR for NH₄⁺ photosynthesis on BaO_NCs-TNS (Fig. 4a and Supplementary Figs. 33 and 34). After 3 h' irradiation, 29.64 mmol $g_{cat}^{-1}$ NO₃⁻ is reduced

to generate 29.26 mmol $g_{cat}^{-1}$ NH₄⁺. Besides, the amount of total N species remains stable in the reaction mixture throughout the entire reaction, which implies that the five-electron transfer for partial reduction of NO₃⁻ to N₂ is effectively impeded. In addition, generated H₂ is detected (Supplementary Fig. 35) since water splitting is the primary side reaction in our reaction system. Trace amounts (2.79 mmol $g_{cat}^{-1}$ for 3 h) of H₂ are produced. By comparing the eight-electron NO₃⁻RR and two-electron water splitting reaction, the selectivity for NH₄⁺ photosynthesis is determined to reach as high as 97.67%. The NH₄⁺ synthesis rate and selectivity of the NO₃⁻RR on BaO_NCs-TNS are superior among the routes under ambient conditions, exceeding that of the other photocatalytic NO₃⁻RR works and even some leading electrocatalytic NO₃⁻RR work (Fig. 4b and Supplementary Table 2). Not surprisingly, the results of this work also exhibit advances in comparison with those of N₂RR routes at ambient conditions, including electrocatalytic, photocatalytic, and photoelectrochemical methods. Subsequently, the reaction activation energy of NO₃⁻-NH₄⁺ synthesis and water splitting for H₂ production was calculated (Fig. 4c and Supplementary Fig. 36). A distinct activation energy decrease of 1.42 eV is noted for the NO₃⁻RR compared with that of the water splitting reaction, which can enable efficient inhibition of electron consumption for the side reaction.

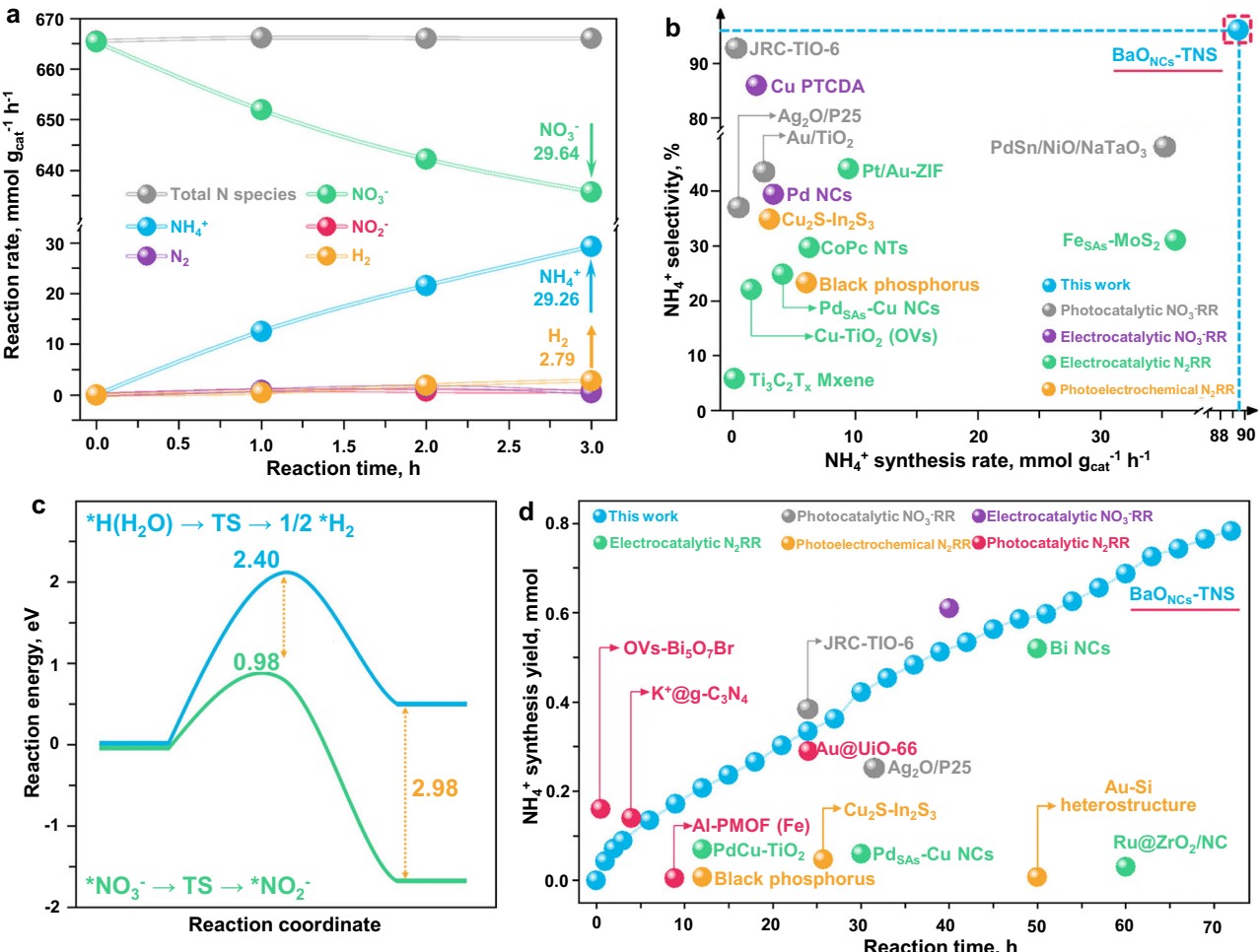

**Fig. 4 Selectivity and long-term stability tests. a** $NH_4^+$ selectivity test from $NO_3^-$RR versus the other potential products. The related stand curves were provided in Supplementary Figs. 37–39. **b** Comparison of $NH_4^+$ production rate and selectivity with different ammonia synthesis routes under ambient conditions[11,16,24,59–69]. **c** Calculated activation energy for $NO_3^-$RR for $NH_4^+$ synthesis and water splitting for $H_2$ generation. **d** Long-term stability of $BaO_{NCs}$-TNS and comparison of total $NH_4^+$ yield with different $NH_4^+$ synthesis routes[11,16,25,42,59,68–75]. The detailed comparison lists of **b** and **d** are provided in Supplementary Table 2.

Despite the ultra-high $NH_4^+$ photosynthesis rate and selectivity, the total $NH_4^+$ yield is a pivotal benchmark to evaluate the performance of ammonia synthesis catalysts and routes; notably, this benchmark is usually overlooked. Hence, long-term tests were performed to determine the accumulation of $NH_4^+$ (Fig. 4d). It is noted that a small amount of deactivation is observed in the first 3 h, which may be caused by the competing reduction reaction of $Ba^{2+}$ and $NO_3^-$. After the $BaO_{NCs}$ are stably formed on TNS, the significant stable production of $NH_4^+$ on the $BaO_{NCs}$-TNS composite is realized. As a result, a total $NH_4^+$ yield of 0.78 mmol is reached within 72 h. Moreover, the catalyst structure is well maintained after the long-term tests (Supplementary Figs. 40–43). By comparing the $NH_4^+$ synthesis efficiency between the $NO_3^-$RR for $NH_4^+$ photosynthesis and recently reported state-of-the-art routes (Supplementary Table 2), the $NH_4^+$ synthesis rate, selectivity, long-term stability and total $NH_4^+$ yield in this work all demonstrate advances in comparison with those of other ammonia synthesis routes under ambient conditions, including electrocatalytic, photocatalytic and photoelectrochemical methods. In addition, in the area of the $NO_3^-$RR for $NH_4^+$ synthesis, the yield of photosynthesis in this work (0.51 mmol within 40 h) is comparable to that of some leading work using electricity as the catalytic driving force (0.61 mmol within 40 h)[11]. Since the density of energy input for

photosynthesis is lower than that of electrocatalysis, the current photosynthesis performance of the $NO_3^-$RR for $NH_4^+$ is very competitive.

**Comprehensive understanding of the reaction mechanism.** To further verify the active radicals responsible for the superior $NO_3^-$RR on the $BaO_{NCs}$-TNS composite, we applied the liquid-state EPR technology using 2,2,6,6-tetramethyl-1-piperidinyloxy (TEMPO) as the trapping reagent (Fig. 5a). The signals of TEMPO decrease rapidly on $BaO_{NCs}$-TNS compared with the TEMPO signals on pristine TNS under light irradiation; this result indicates that more photoexcited electrons are generated and consumed on $BaO_{NCs}$-TNS. The incorporation of $H^+$ and TEMPO-$e^-$ is significantly strengthened via the *operando* construction of subnanometric $BaO_{NCs}$, generating abundant active protons ($H^*$) to catalyze the $NO_3^-$RR for $NH_4^+$ photosynthesis. In addition, enhanced •OH, •$O_2^-$ and $^1O_2$ production is also observed on $BaO_{NCs}$-TNS (Supplementary Fig. 44). These reactive oxygen species (ROS) are beneficial for the photocatalytic oxidation of EG, which facilitates hole consumption and charge separation.

Moreover, an in situ diffused reflectance infrared Fourier transform spectroscopy (DRIFTS) technique was introduced to dynamically detect the primary reaction process of the $NO_3^-$RR. As

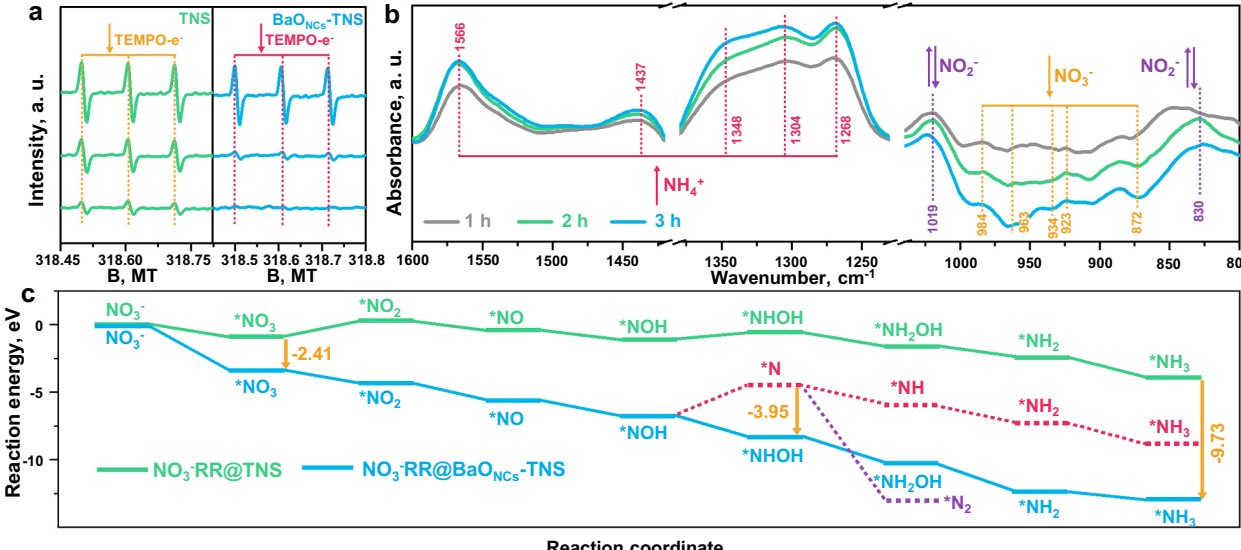

**Fig. 5 Molecular-level reaction mechanism of $NO_3^-$RR for $NH_3$ photosynthesis. a** EPR spectra for 2,2,6,6-tetramethyl-1-piperidinyloxy (TEMPO)-e$^-$ showing the reactive species. **b** In situ diffused reflectance infrared Fourier transform spectroscopy (DRIFTS) revealing the principal reactants and products within the $NO_3^-$RR on BaO$_{NCs}$-TNS. **c** Gibbs free-energy diagram of the reaction coordinates. Steps marked red and purple are the potential side reactions of $NO_3^-$RR on BaO$_{NCs}$-TNS.

shown in Fig. 4b, the characteristic peaks regarding adsorbed $NO_3^-$ are clearly identified at 872, 923, 934, 969, and 984 cm$^{-1}$ [38,39]. It is noted that the intensity of $NO_3^-$ gradually weakens with prolonged irradiation time, which confirms the consumption and reduction of $NO_3^-$. As the $NO_3^-$RR proceeds, the IR signals of the intermediate $NO_2^-$ are observed at 830 and 1019 cm$^{-1}$ [40,41] and increase within the first two hours. After that, a decrease in the $NO_2^-$ signals are noted (blue line in Fig. 4b), which illustrates that more $NO_2^-$ has been consumed than generated. Most importantly, the continuous production of $NH_4^+$ is verified (1268, 1304, 1348, 1437, and 1566 cm$^{-1}$) [42–44]. These results clarify that the $NO_3^-$RR route for $NH_4^+$ photosynthesis is feasible, in which $NO_2^-$ is identified as the principal intermediate product. Based on the in situ DRIFTS detection, the primary reaction pathways of the $NO_3^-$RR for $NH_4^+$ photosynthesis are summarized (Note 1).

**Note 1** Primary reaction pathways of $NO_3^-$ reduction for $NH_4^+$ photosynthesis

$$NO_3^- + H^+ \rightarrow {}^*NO_3 + {}^*H \quad (1)$$

$${}^*NO_3 + {}^\bullet H \rightarrow {}^*NO_2 + {}^*OH \quad (2)$$

$${}^*NO_2 + {}^\bullet H \rightarrow {}^*NO + {}^*OH \quad (3)$$

$${}^*NO + {}^\bullet H \rightarrow {}^*NOH \quad (4)$$

$${}^*NOH + {}^\bullet H \rightarrow {}^*NHOH \quad (5)$$

$${}^*NHOH + {}^\bullet H \rightarrow {}^*NH_2OH \quad (6)$$

$${}^*NH_2OH + {}^\bullet H \rightarrow {}^*NH_2 + H_2O \quad (7)$$

$${}^*NH_2 + {}^\bullet H \rightarrow {}^*NH_3 \quad (8)$$

The activation and reduction of $NO_3^-$ on the catalyst surface were subsequently revealed by DFT calculations to support the experimental results. As depicted in Supplementary Fig. 45, the adsorption energy and received electrons of $NO_3^-$ at the BaO$_{NCs}$/TNS interface is observably elevated compared with that of pristine TNS, which could promote the $NO_3^-$ reduction process.

Furthermore, the Gibbs free-energy diagrams ($\Delta G$) were obtained to verify the effect of subnanometric BaO$_{NCs}$ construction on the reaction energy and pathways (Fig. 5c). Referring to the experimental results, we first compared the eight-electron transfer reaction for the synthesis of $NO_3^-$–$NH_3$ on pristine TNS (green line in Fig. 5c and Supplementary Fig. 46) and BaO$_{NCs}$-TNS (blue line and Supplementary Fig. 47). It is clearly revealed that facile $NO_3^-$ dissociation (${}^*NO_3$-${}^*NO_2$) can be accomplished on BaO$_{NCs}$-TNS with an observable decrease in energy compared to that of pristine TNS, which is the dominant reason for the elevated $NO_3^-$RR performance on BaO$_{NCs}$-TNS. As the $NO_3^-$RR proceeds, a total energy decrease of 9.73 eV is noted for efficient $NH_3$ synthesis. Two primary competing reaction pathways regarding $N_2$ and $NH_4^+$ production were compared (Supplementary Note 2 and 3). A lower $\Delta G$ is required for ${}^*NOH$-${}^*NHOH$ (−1.63 eV) reduction than for ${}^*NOH$-${}^*N$ (+2.31 eV) reduction (red line in Fig. 5c and Supplementary Fig. 48). Since ${}^*N$ generation is prevented, $N_2$ production (purple line in Fig. 5c and Supplementary Fig. 49) is not an optional process on BaO$_{NCs}$-TNS. Besides, the lower activation energy for $NO_3^-$ dissociation (0.98 eV, Fig. 4c) is identified in comparison with that of the water splitting reaction (2.40 eV). Hence, among the three potential products in these reaction pathways ($NH_4^+$, $N_2$ and $H_2$), the highly selective eight-electron reduction of $NO_3^-$ for $NH_4^+$ photosynthesis is achieved via the assistance of MO$_{NCs}$.

**Practical applications of $NO_3^-$RR in simulated wastewater.** The practical application of the as-proposed $NO_3^-$RR for ammonia photosynthesis route was developed. Since the organic matter of EG is applied in the catalysis system (Cat. Sys.) as the hole sacrificial agent, the conversion pathways of EG are investigated via the in situ DRIFTS technology (Fig. 6a). It is observed that the dynamic adsorption equilibrium (Ads. Equil.) of EG is gradually formed based on the detection of methane (2940, 2880, 1437, and 1364 cm$^{-1}$) [45,46] and alcohol (1123, 1080, and 1040 cm$^{-1}$) [45,47] species. Then the generation and accumulation of formate (1153 cm$^{-1}$) [48] and carbonate (1285 cm$^{-1}$) [46,47] are observed, which can be attributed to the primary products for EG oxidation. Hence, it is concluded that the reactions of EG oxidation and $NO_3^-$ reduction proceed

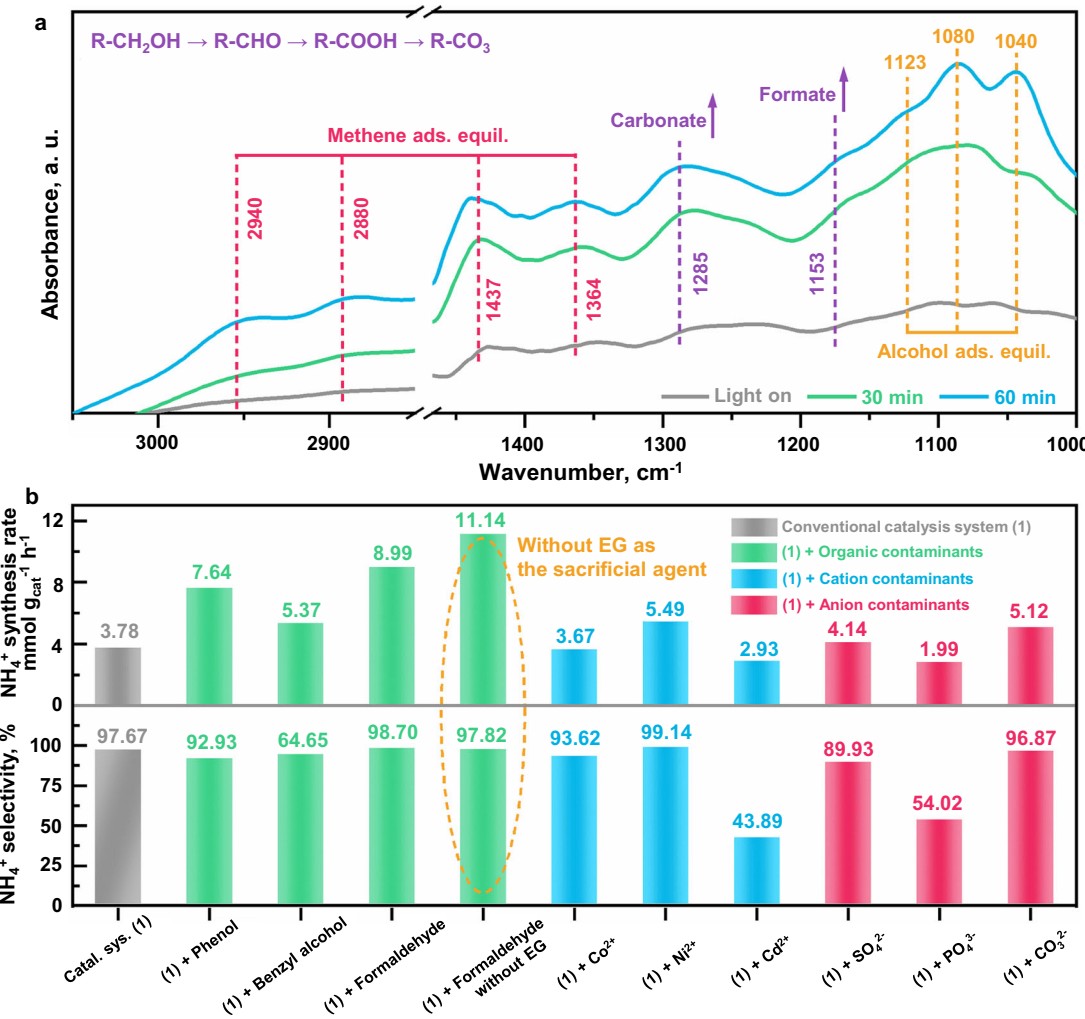

**Fig. 6 Practical application of NO₃⁻RR to NH₄⁺ route in simulated wastewater. a** In situ DRIFTS for ethylene glycol (EG, hole sacrificial agent) oxidation during the NO₃⁻RR. **b** Ammonia synthesis rates and selectivity evaluation by adding different types of simulated wastewater into the catalysis system (Catal. Sys.), including the organic matter (phenol, benzyl alcohol and formaldehyde), cation ($Co^{2+}$, $Ni^{2+}$ and $Cd^{2+}$) and anion ($SO_4^{2-}$, $PO_4^{3-}$ and $CO_3^{2-}$) contaminants correspondingly. As for the condition of formaldehyde, the catalytic tests were conducted with and without EG respectively.

simultaneously, in which the hole consumption by EG oxidation could in turn accelerate the NO₃⁻RR to promote ammonia synthesis.

Most importantly, it should be noted that abundant organic contaminants are distributed in many $NO_3^-$-containing emission conditions such as agricultural and chemical wastewater degradation and drinking water purification, in which the organic contaminants can be utilized as what is called the hole sacrificial agent. Based on this consideration, phenol, benzyl alcohol and formaldehyde were added into the catal. sys. as potential contaminants respectively (Fig. 6b and Supplementary Fig. 50a–f). It is found that the ammonia synthesis rates and selectivity are all retained, which indicates that the NO₃⁻RR route is established in the simulated organic wastewater. Interestingly, the ammonia synthesis rates are increased in the order of conventional catal. sys. ($3.78\ mmol\ g^{-1}\ h^{-1}$) < benzyl alcohol ($5.37\ mmol\ g^{-1}\ h^{-1}$) < phenol ($7.64\ mmol\ g^{-1}\ h^{-1}$) < formaldehyde ($8.99\ mmol\ g^{-1}\ h^{-1}$). It is deduced that the hole consumption capacity of these organic contaminants is higher than that of EG, which leaves more electrons to catalyze the NO₃⁻RR. Then, the corresponding test of formaldehyde was conducted without EG as the hole sacrificial agent (Fig. 6b and Supplementary Fig. 50g–h). It is observed that the ammonia synthesis rate is further elevated to $11.14\ mmol\ g^{-1}\ h^{-1}$ with 97.82% of selectivity noted, which reveals that the formaldehyde can act as the "hole sacrificial agent" more

efficiently than that of EG. Based on the organic contaminant investigation in the simulated wastewater, it is clarified that the NO₃⁻RR route can be developed as a "sacrificial-agent-free" technology for the application of ammonia synthesis in wastewater coupled with the organic pollutants' oxidation, which demonstrates scientific significance in both areas of environmental remediation and energy conversion. Besides, the addition of cation contaminants ($Co^{2+}$, $Ni^{2+}$, and $Cd^{2+}$, Supplementary Fig. 51) and anion contaminants ($SO_4^{2+}$, $PO_4^{3-}$ and $CO_3^{2-}$, Supplementary Fig. 52) have also been considered, in which the catalytic efficiency is accomplished in general. Some discrepancy in the performance is noted among these ions, which could be raised by the complicated impact of added ions on the catal. sys. and requires further investigation in the future.

## Discussion

In summary, a highly active and selective NO₃⁻RR for NH₄⁺ photosynthesis was achieved by *operando* construction of sub-nanometric $MO_{NCs}$ on TNS. The dynamic evolution pattern, growth mechanism, and interfacial structure of $MO_{NCs}$ were characterized and well-defined. A superior NH₄⁺ photosynthesis rate, selectivity, long-term stability and total yield were achieved among the various NH₄⁺ synthesis routes under ambient

conditions. Then it was proposed that the unique electronic structure at the $MO_{NCs}$/TNS interface was mainly responsible for the enhanced $NO_3^-$ dissociation and eight-electron reduction reaction. The practical application of $NO_3^-RR$ route in simulated wastewater was developed, which demonstrated scientific significance in both areas of environmental remediation and energy conversion. The discovery of subnanometric $MO_{NCs}$ for sustainable $NO_3^-$-$NH_4^+$ photosynthesis is inspiring and is of general knowledge, thereby providing numerous opportunities for research into cluster chemistry and artificial photosynthesis.

## Methods

**Chemicals.** All chemicals were purchased without further treatment. The respective source and purity were listed in Supplementary Table 3.

**Synthesis of TNS.** The synthesis of $TiO_2$ nanosheets (TNS) was conducted via slight modification of the reported work[49,50]. In a typical preparation procedure, 3 mL of HF solution was dropped slowly into 25 mL of $Ti(OBu)_4$•(TBOT) under continuous magnetic stirring for 2 h until the solution formed into a gel-like solution. Then the mixture was transferred into a Teflon autoclave with a volume of 50 mL and then heated at 180 °C for 24 h. After naturally cooling down to room temperature, the powder was separated and collected by high-speed centrifugation at 10,000 rpm for 5 min with distilled water (DI) and ethanol washing for at least 10 times respectively, which removes the residual organic matter and $F^-$. At last, the obtained sample was dried at 80 °C overnight in the vacuum drying oven.

**Construction of $MO_{NCs}$ on TNS.** Five milligrams of TNS powder was dispersed in 100 mL of reaction solution containing 10 mg/L of $NO_3^-$-N, 50 mg/L of alkaline earth chlorides ($MgCl_2$, $CaCl_2$, $SrCl_2$, and $BaCl_2$ respectively) and 10 mL EG. Then the mixture was transferred into a photocatalytic reactor (MC-GF250, Merry Change, China) and degassed for soluble air with high-purity Ar (99.999%) at 50 mL/min for 30 min under stirring. A 300 W Xe lamp (MC-X301B, Merry Change, China) was used at the light source. After 1 h's irradiation with continuous Ar pumping and stirring, the obtained powder was collected by washing with ethanol and DI for three times respectively. With the irradiation on stream, the reaction mixture was extracted several times for the tests of $M^{X+}$ concentration and *operando* evolution of $MO_{NCs}$ on the substrate surface.

**Catalyst characterization.** The crystal information was examined by the X-ray diffraction (XRD, model D/max RA, Rigaku Co.) technology. The morphology was surveyed by scanning electron microscopy (SEM, XL30 ESEM FEG), transmission electron microscopy (TEM, FEI Talos F200S). The *operando* evolution of $MO_{NCs}$ was verified by quasi in situ high-angle annular dark-field scanning transmission electron microscopy (HAADF-STEM, JEOL JEM-ARM200F) with spherical aberration correction to investigate the morphology at the atomic scale. The chemical composition was tested via the X-ray photoelectron spectroscopy (XPS, Thermo Scientific K-Alpha plus) with an Al Kα X-ray light source. The elemental component analysis was conducted by X-ray fluorescence (XRF, BRUKER, M4 TORNADO). The solid-state EPR (JES FA200) spectra were performed to identify the vacancy signals.

**Optical and electronic property identification.** The light absorption capacity was performed by a scanning UV–vis spectrophotometer (Shimadzu UV-2450) outfitted with an integrating sphere, using barium sulfate as the comparison sample. The Mott-Schottky spectra were conducted using catalysts/C, Ag/AgCl and Pt as working, reference and counter electrodes respectively on an electrochemical workstation (CHI-660E), and the results were recorded from −1.0 to 1.5 V at 1000 Hz without light irradiation. The steady photoluminescence (PL) spectra were investigated using a fluorescence spectrophotometer (Edinburgh Instruments FLSP-920). Time-resolved fluorescence emission decay spectra (PicoQuant Fluotime 300) were carried out to verify the carrier's life time under light irradiation. The liquid-state EPR spectra of active radicals were obtained on a JES FA200 spectrometer to investigate the production of the ROS under light illumination. The 5, 5-Dimethyl-1-Pyrroline-N-Oxide (DMPO) was used as the trapping agent to confirm the involvement of DMPO-•OH and DMPO-•$O_2^-$ respectively in aqueous methanol dispersion. 4-oxo-2, 2, 6, 6-Tetramethyl-1-Piperidinyloxy (TEMP) was applied to survey the TEMP-$^1O_2$ and TEMP-1-oxyl (TEMPO) was used to characterize the photo-induced electrons (TEMPO-$e^-$).

**DFT calculation.** The spin-polarized DFT calculations were operated with the "Vienna ab initio simulation package" (VASP 5.4), in which the PBE exchange-correlation functional was included[51–53]. The PBE + U correction (U = 4.0 eV) was implemented to account for the on-site charge interaction of the d electrons in Ti elements[54], which improved the accuracy for the calculations of electron migration at the $MO_{NCs}$-TNS interfaces. The cut-off energy was set to 400 eV. K points in the Brillouin zone were set to 5 × 5 × 1 for both structural and electronic optimization.

Geometry relaxation was achieved after the residual forces were smaller than 0.01 eV Å$^{-1}$. The Gaussian smearing width was set to 0.2 eV. The initial calculation model of TNS contains 60 Ti atoms and 120 O atoms respectively (Supplementary Fig. 12a). The typical [001] facet is exposed for further calculations. The lattice parameters were set to 11 × 15 × 25 Å$^3$, which contains a vacuum slab of 15 Å to impede the potential interaction between neighboring lattices. The initial calculation models of $MgO_{NCs}$, $CaO_{NCs}$, $SrO_{NCs}$ and $BaO_{NCs}$ were constructed with the cluster size of 0.73, 0.83, 0.89, and 0.96 nm respectively (Supplementary Fig. 12b–e).

The adsorption energy ($E_{ads}$) for molecules was calculated as follows:

$$E_{ads} = E_{tot} - E_{cat.} - E_{mol} \qquad (9)$$

where $E_{tot}$, $E_{cat.}$, and $E_{mol}$ depicted the total energy of adsorption structure, catalyst support, and isolated molecule respectively.

The Gibbs free energy variation ($\Delta G$)[55,56] between the initial state (IS) and final state (FS) was determined with the following equation

$$\Delta G = E_{FS} - E_{IS} + \Delta E_{ZPE} - T\Delta S \qquad (10)$$

where $E_{FS}$ and $E_{IS}$ referred to the DFT total energy for FS and IS correspondingly. $\Delta E_{ZPE}$ and $\Delta S$ denoted the variation of zero-point energy and entropy. The room temperature ($T$, 298.15 K) was applied.

The climbing image-nudged elastic band (CI-NEB)[57,58] code was conducted to identify the reaction coordinates from IS to FS, which located the transition state (TS) with single imaginary frequency verification. The activation energy ($E_a$) and reaction energy ($E_r$) were defined as follows

$$E_a = E_{TS} - E_{IS} \qquad (11)$$

$$E_r = E_{FS} - E_{IS} \qquad (12)$$

where $E_{IS}$, $E_{TS}$, and $E_{FS}$ were DFT calculated total energy of IS, TS, and FS respectively.

**$NO_3^-$-$NH_4^+$ photosynthesis efficiency test.** Photosynthesis tests were first conducted to determine the activity enhancement by operando $MO_{NCs}$ construction. In a typical experimental procedure, 5 mg of TNS powder was dispersed in 100 mL of reaction solution containing 10 mg/L of $NO_3^-$-N, 50 mg/L of alkaline earth chlorides ($MgCl_2$, $CaCl_2$, $SrCl_2$, and $BaCl_2$ respectively) and 10 mL EG. Then the mixture was transferred into a photocatalytic reactor (MC-GF250, Merry Change, China) and degassed for soluble air with Ar at 50 mL/min for 30 min under stirring. A 300 W Xe lamp (MC-X301B, Merry Change, China) was used at the light source. After the photocatalysis reaction, the *operando* construction of $MO_{NCs}$ on TNS is accomplished. The photocatalysts were collected and washed for further characterization. The blank control experiment was also performed, which excluded the catalysts and $NO_3^-$-N respectively. The consumed $NO_3^-$ and produced $NH_4^+$ were both detected by ion chromatography (IC, Shimadzu IC-16 for $NH_4^+$ and Analysis Lab CS2000 for $NO_3^-$ respectively). After that, the reaction parameter was comprehensively optimized to obtain the optimal $NH_4^+$ photosynthesis efficiency, which included $NO_3^-$-N concentration (from 10 to 500 mg/L), catalyst dosage (from 0.5 to 5 mg) and the irradiation source (UV, full spectrum and simulated solar light). The pH of this catalytic system remains at ca. 7.0 during the test since the $KNO_3$ and EG consist of neutral solutions (Supplementary Fig. 53). The temperature is controlled at 25 °C by using the circulating chiller (Supplementary Fig. 54).

The quantitative $^{15}N$ isotope tracing measurement was conducted, using 10 mg/L of $K^{14}NO_3$ and $K^{15}NO_3$ as the feedstock respectively, the produced $^{14}NH_4^+$ and $^{15}NH_4^+$ were quantified by IC. Then the $^1H$ NMR (Bruker 400 M) was used to complement the IC results. As for the long-term stability test, the $NO_3^-$-N concentration was increased to 500 mg/L to guarantee sufficient feedstock. Meanwhile, the catalyst dosage was increased to 50 mg to elevate the total yield of $NH_4^+$. In order to investigate the $NH_4^+$ selectivity, the $H_2$ was also detected during the $NO_3^-RR$, using gas chromatography (Kechuang GC 2002) with the thermal conductivity detector (TCD). The selectivity was calculated as follows

$$NH_4^+ \text{ Selectivity} = [8 \text{ Yield} (NH_4^+)]/[2 \text{ Yield} (H_2) + 5 \text{ Yield} (1/2\ N_2) + 8 \text{ Yield} (NH_4^+)] \times 100\% \qquad (13)$$

**In situ DRIFTS investigation.** In situ diffuse reflectance infrared Fourier transform spectroscopy (DRIFTS, Bruker INVENIO R) was utilized to monitor the adsorbed species on the $BaO_{NCs}$-TNS surface within the reaction process, in which an in situ diffuse-reflectance cell (Harrick) and a reaction chamber (HVC) were equipped. Before measurement, the catalyst was mixed into 100 mg/L of $NO_3^-$-N solution by continuous stirring and then dried at 110 °C in a vacuum drying oven. The as-prepared sample was transferred in the reaction chamber and heated for 30 min at 110 °C to completely remove the adsorbed species on the surface. High-purity He (99.999%) was continuously pumped into the reaction system to maintain the inert atmosphere. A Xe lamp was used as the light source. The IR detection was conducted during the light irradiation.

**Practical applications in simulated wastewater.** Some potential contaminants were added into the catalysis system of $BaO_{NCs}$-TNS to evaluate the $NO_3^-RR$

performance in simulated wastewater, including phenol, benzyl alcohol, and formaldehyde as organic matters, $Co^{2+}$, $Ni^{2+}$, and $Cd^{2+}$ as cations, $SO_4^{2-}$, $PO_4^{3-}$ and $CO_3^{2-}$ as anions respectively. $CoCl_2$, $NiCl_2$, and $CdCl_2$ were used as the source of cations. $K_2SO_4$, $K_3PO_4$, and $K_2CO_3$ were used as the source of anions respectively. Based on the formaldehyde test, the EG is excluded from the reaction mixture for comparison. The concentration variation of $SO_4^{2-}$, $PO_4^{3-}$, and $CO_3^{2-}$ is determined by ion chromatograph (Shimadzu IC-16). An inductive coupled plasma emission spectrometer (Agilent ICPOES730) is applied to reveal the concentration of $Co^{2+}$, $Ni^{2+}$, and $Cd^{2+}$. Phenol and benzyl alcohol were tested on the liquid chromatography (Shimadzu LC-20AT). In addition, we measured the concentration of formic acid as the oxidative product of formaldehyde by IC (Shimadzu IC-16).

## Data availability

All data generated in this study are provided in the Source Data files, in which the data presented in Figures from the Maintext and Supplementary Information are listed in the Excel files of Source Data 1 and Source Data 2 respectively.

## Code availability

Only the commercial codes were used in this work (See references).

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

## Acknowledgements

This work was supported by the National key R&D project of China (Grant No. 2020YFA0710000), National Natural Science Foundation of China (Grant Nos. 21822601, 22176029, and 22006009), Excellent Youth Foundation of Sichuan Scientific Committee Grant in China (No. 2021JDJQ0006), the Fundamental Research Funds for the Central Universities (ZYGX2019Z021) and the 111 Project (B20030).

## Author contributions

J.Y.L and F.D. conceived and designed the experiments. J.Y.L., R.M.C., and J.L.W carried out the materials fabrication, performance tests, characterizations, and calculations. Y.Z. and G.D.Y. helped analyzed the experimental data. J.Y.L. and F.D. wrote the paper, and all authors discussed the results and commented on the manuscript.

## Competing interests

The authors declare no competing interests.
