## [Peer review file · Nature Communications]

Reviewers' comments:

Reviewer #1 (Remarks to the Author):

Recommendation: Reconsider after major revisions.

Comments:

The current manuscript treats the timeliest topic of NO₃-RR by operando construction of subnanometric MONCs on TNS using photocatalysis at ambient conditions. The manuscript is well structured, and the results are properly presented and generally convincing. A superior NH₄⁺ photosynthesis rate, selectivity, long-term stability, and total yield were achieved. However, the scientific significance and novelty of reducing nitrites (most of which are produced by oxidizing ammonia) to ammonia is in doubt. The manuscript could not be published unless several key issues are fully resolved by the authors. The major concerns include:

1. The novelty of the subnanometric alkaline-earth oxide clusters used in NO₃-RR should be highlighted more. The comparisons with other alkaline-earth-containing N₂RR/NO₃-RR catalysts should be presented. In addition, several latest references (e.g., Nat. Commun. 2021, 12, 2870) should be added in this context.
2. The authors must clarify the significance to reducing nitrite to ammonia, given the fact that most industrial/commercial nitrite (and even those in wastewater) were produced by oxidizing ammonia (the Ostwald process). Therefore, consuming additional energy to reduce such "nitrate" back to ammonia seems like to be wasted efforts (especially when using ultrapure chemicals as nitrate sources). In addition, if the authors could show that the nitrates in wastewater could be selectively turned into ammonia, this work would be much more convincing and significant. Otherwise, the better usage of collected and purified environmental nitrites is to produce nitric acid or nitrite fertilizer instead of reducing them back to ammonia.
3. The author claimed that the construction of MONCs and the improvement of NH₄⁺ photosynthesis rate were completed at the same time. If it is done simultaneously, the ammonia production rate (increasing slope) should show a gradual increase as the reaction time progresses. However, Fig. 3a shows that the catalyst maintains a stable ammonia production rate (constant slope). I suggest clarifying that point. Or, what's the timeline for the formation of MONCs on TNS?
4. The authors may need to explain the y-axis for Fig. 3c,d.
5. In the photocatalytic reaction operation method described by the author (Lines 415-430), clusters are grown on TNS in the catalytic reaction. Would the catalyst need to be washed? Nevertheless, the catalyst was washed in the catalyst synthesis (Line 356). This part needs further clarity.
6. As shown in Fig 3a, TNS itself exhibited nice catalytic activity. The active sites in TNS and MONCs-TNS should be elucidated more clearly in the manuscript.
7. In Fig 4c, the energy of the transition state is not marked. Then, how to calculate the energy difference of 1.42 eV?
8. The author needs to calculate band gaps and band edges through Mott-Schottky, XPS valence band spectrum, diffuse reflectance spectrum, etc.
9. Further characterization like XAFS of BaSAs and BaONCs should be supplemented to unambiguously describe a more realistic structure of active sites.
10. In Supplementary Figs. 5 and 6, the elemental distribution of Mg and Ca (esp. Ca) were not overlapped with TNS. Why? Do these isolated MO clusters exhibit catalytic activity? The authors may need to clarify this point.
11. The direct comparison in selectivity/activity for NO₃-RR and N₂RR is unfair and unnecessary. The scientific significance and the difficulty for the activation of N-N triple bond (nitrogen fixation) is much higher than just reducing nitrite (most of which comes from ammonia) back to ammonia.

Reviewer #2 (Remarks to the Author):

The manuscript describes that the BaO_x nanoclusters loaded on TiO₂ nanosheets promote

photocatalytic NO₃⁻ to NH₄⁺ reduction in water with sacrificial electron donor under UV irradiation. I cannot recommend the manuscript for publication. Followings should be considered.

1) Almost 100% selectivity for NO₃⁻ to NH₄⁺ on bare TiO₂ has already been reported (Hirakawa, H.; Hashimoto, M.; Shiraishi, Y.; Hirai, T. Selective Nitrate-to-Ammonia Transformation on Surface Defects of Titanium Dioxide Photocatalysts. *ACS Catal.* 2017, 7 (5), 3713–3720.). This report was not cited in the manuscript. The results presented in the present manuscript are therefore not surprising.

2) The authors claimed that the BaOx nanoclusters behave as active sites for NO₃⁻ reduction only by DFT calculations. This is very weak to support the assumptions provided here. In the present system, many Cl⁻ ions present in the solution (see the experimental section). The possible mechanism for the reaction enhancement by the addition of BaCl₂ to the solution is that Cl⁻ behaves as a sacrificial electron donor (see: Huang, L.; Li, R.; Chong, R.; Liu, G.; Han, J.; Li, C. Cl⁻ Making Overall Water Splitting Possible on TiO₂-Based Photocatalysts. *Catal. Sci. Technol.* 2014, 4 (9), 2913–2918).

Based on the above, I do not think the novelty and quality of this manuscript justifies publication in *Nat. Commun.*

Reviewer #3 (Remarks to the Author):

Review attached

A highly active and selective nitrate to NH_4^+ photocatalysis was achieved by operando construction of subnanometric alkaline-earth oxide clusters on TiO_2 nanosheets. The optimal ammonia synthesis rate and total ammonia synthesis yield are promising compared to other reported systems (Table S1). The authors support their findings with a plethora of experimental and computational techniques (XRD, SEM, TEM, STEM, elemental mapping, XPS, IC, XRF, EPR, DFT calculations, UV-VIS, FTIR, NMR, ESR, XRF, isotope labeled studied). The analysis and figures are of high technical quality. The manuscript is quite well-written. The supporting information contains many useful details, including blank experiments. This work should be of significance to the field and related fields and appears original. The work supports the conclusions and claims. The methodology seems sound and meets the expected standards in the field. The work seems publishable with minor revisions.

Technical Comments

(1) DFT calculations using the standard PBE functional predicted electron migration from the BaO_{NCs} to TNS. If the more accurate $\text{PBE}+U$ or a hybrid functional is used, does the direction of electron migration remain the same? In systems with d- and f- localized electrons the overestimation of electron delocalization is a known weakness of PBE.

(2) What was the pH of the system? I did not see it stated. Does the pH of the system change with time?

Minor Comments

(3) Were all measurements done at room temperature? I did not see this clearly stated.

(4) **Abstract:** The authors wrote: “A total ammonia yield of 0.78 mmol within 72 hours is achieved, which is even superior to some electro-driven N_2 to ammonia synthesis routes under ambient conditions.”

➔ This does not seem to be a fair comparison, as electro-driven N_2 to ammonia activity under ambient conditions is very low and a different reaction. It would be a fairer comparison to contrast against electro-driven nitrate to ammonia synthesis routes.

(5) The authors wrote “The catalyst exhibits an ultrahigh ammonia photosynthesis rate of $11.97 \text{ mol g}_{\text{metal}}^{-1} \text{ h}^{-1}$ ($89.79 \text{ mmol g}_{\text{cat}}^{-1} \text{ h}^{-1}$) with nearly 100% selectivity.”

➔ How is there more g_{cat} than g_{metal} by almost 3 orders of magnitude? Is Ti of the TiO_2 not included as part of g_{metal} ?

(6) **Page 4:** “Key challenge lines in revealing the interfacial structure of active sites for facilitating eight-electron transfer”

➔ I believe “lines” should be “lies”.

(7) **Figure 1c/d.** The inset count labels are not readable unless zoom in to 170%. Perhaps there is a way to make the size distribution image and labels slightly larger.

(8) “The deconvolution of the Ba $3d$ XPS spectrum illustrates a perfect fitting to four characteristic peaks at binding energies of 794.61, 792.84, 778.99 and 777.15 eV.”

→ Is the fitting really “perfect”? Perhaps vary the word choice.

(9) “**Supplementary Figure 8.** Variation of IC signals for Mg^{2+} (a), Ca^{2+} (b) and Sr^{2+} (c) ions during the operando construction of corresponding MO_{NCs} on TNTs.”

→ “On” should not be subscripted. Do the authors mean “TNSs” instead of “TNTs”?

Response to Referee 1:

Recommendation: Reconsider after major revisions.

General Comment: The current manuscript treats the timeliest topic of NO_3^- RR by operando construction of subnanometric MO_{NCs} on TNS using photocatalysis at ambient conditions. The manuscript is well structured, and the results are properly presented and generally convincing. A superior NH_4^+ photosynthesis rate, selectivity, long-term stability, and total yield were achieved. However, the scientific significance and novelty of reducing nitrites (most of which are produced by oxidizing ammonia) to ammonia is in doubt. The manuscript could not be published unless several key issues are fully resolved by the authors.

Response: Thanks for the patience and carefulness of the Reviewer. The raised comments and suggestions are very helpful to improve this manuscript. The detailed comments are responded point by point, and we carefully revised the Manuscript (MS) and Supplementary Information (SI) files according to the Reviewer's comments.

As for the significance and novelty issues of the nitrate reduction routes, we conducted additional experiments by using simulated wastewater as the nitrate source, which indicates that the nitrate-ammonia route can be developed for practical application in general wastewater occasion (*refer to Comment 2 for detailed description*). The significance to reducing nitrite to ammonia is also clarified in the *Comment 2*. We believe that the new understanding of cluster catalysis, ammonia synthesis and nitrogen-cycle utilization is provided by our findings in the work, which shows scientific significance in the both area of environmental remediation and energy conversion.

Comment 1: The novelty of the subnanometric alkaline-earth oxide clusters used in NO_3^- RR should be highlighted more. The comparisons with other alkaline-earth-containing N_2 RR/ NO_3^- RR catalysts should be presented. In addition, several latest references (e.g., *Nat. Commun.* 2021, 12, 2870) should be added in this context.

Response: Thanks for the suggestion. The novelty of the subnanometric alkaline-earth oxide clusters used in NO_3^- RR is highlighted in the revised MS. Then the comparison between BaO_{NCs} -TNS and the other alkaline-earth-containing catalysts has been appended in Supplementary Table 1. Since the reported investigation of alkaline-earth-containing catalysts for ammonia synthesis under ambient condition is fairly few, we added some works applying Haber–Bosch process for comparison. It is concluded that the NO_3^- RR to ammonia photosynthesis manifests superior advance among these routes. Besides, some latest references reporting NO_3^- RR to ammonia synthesis were cited.

●**Page 4 in MS:** Here, we demonstrate a general strategy to accomplish the *operando* construction of subnanometric alkaline-earth oxide clusters (MO_{NCs} , M=Mg, Ca, Sr or Ba) as the active sites on TiO_2 nanosheets (TNS) due to the nontoxicity and low price of alkaline-earth metals. After the *in situ* construction of the MO_{NCs} -TNS composites, highly active and selective NH_4^+ photosynthesis from NO_3^- RR is achieved.

●**Page 10 in MS:** The optimal rate catalyzed by BaO_{NCs} -TNS manifests superior advances in comparison with that of other ammonia synthesis by using alkaline-earth-containing catalysts (Supplementary Table 1).

●**Page 37 in SI:**

Supplementary Table 1 Comparison of the ammonia synthesis efficiency between NO_3^- RR to ammonia photosynthesis catalyzed by BaO_{NCs} in this work and recently reported alkaline earth-based catalysts for ammonia synthesis.

Ammonia synthesis route	Catalyst	Optimal ammonia synthesis rate ($\text{mmol g}_{\text{cat}}^{-1} \text{h}^{-1}$)	Reference
NO_3^-RR to ammonia Photosynthesis	BaO_{NCs}-TNS	89.79	This work
Photocatalytic N_2 RR to ammonia	BMOF(Sr)–0.2Fe	0.78	ACS Catal. , 2021 , 11 , 9986-9995.
Photocatalytic N_2 RR to ammonia	g- C_3N_4 / $\text{Mg}_{1.1}\text{Al}_{0.3}\text{Fe}_{0.2}\text{O}_{1.7}$	0.27	RSC Adv. , 2021 , 7 , 18099-18107.
Haber–Bosch process $T = 360\text{ }^\circ\text{C}$ $P = 0.9\text{ MPa}$	Ru/Ba-Ca(NH_2) ₂	60.4	Angew. Chem. Int. Ed. , 2018 , 57 , 2648-2652.
Haber–Bosch process	Ba–Ru/CNTs-D	49.0	J. Energy Chem. , 2020 , 41 ,

$T = 400\text{ }^{\circ}\text{C}$ $P = 10.0\text{ MPa}$			79-86.
Haber–Bosch process $T = 340\text{ }^{\circ}\text{C}$ $P = 1.0\text{ MPa}$	Ru/BaO-CaH ₂	10.5	ACS Catal. , 2018 , 8 , 10977-10984.
Haber–Bosch process $T = 400\text{ }^{\circ}\text{C}$ $P = 0.1\text{ MPa}$	Ru/Sr ₂ Nb ₂ O ₇	4.98	Appl. Catal. A: Gen. , 2018 , 554, 1-9.
Haber–Bosch process $T = 400\text{ }^{\circ}\text{C}$ $P = 0.1\text{ MPa}$	Cs- Ru/Sr ₂ Ta ₂ O ₇	5.04	J. Catal. , 2020 , 389 , 556-565.
Haber–Bosch process $T = 350\text{ }^{\circ}\text{C}$ $P = 0.9\text{ MPa}$	Ru/Ca(NH ₂) ₂	31.7	ACS Catal. , 2016 , 6 , 7577-7584.
Haber–Bosch process $T = 340\text{ }^{\circ}\text{C}$ $P = 0.1\text{ MPa}$	Ru-Cs/MgO	1.65	ACS Catal. , 2014 , 4 , 674-680.
Haber–Bosch process $T = 300\text{ }^{\circ}\text{C}$ $P = 1.0\text{ MPa}$	LiH/Co-Mg-O-co	19	Chem. Comm. , 2021 , 574 , 8576-8579.
Haber–Bosch process $T = 400\text{ }^{\circ}\text{C}$ $P = 1.0\text{ MPa}$	Ba–Ru/MgO	25.2	J. Catal. , 2003 , 214 , 327-335.

●Page 22 in MS, the following references are cited:

13 Wu, Z. Y. *et al.* Electrochemical ammonia synthesis via nitrate reduction on Fe single atom catalyst. *Nat. Commun.* **12**, 2870 (2021).

18 Kani, N. C. *et al.* Solar-driven electrochemical synthesis of ammonia using nitrate with 11% solar-to-fuel efficiency at ambient conditions. *Energy Environ. Sci*, doi:10.1039/d1ee01879e (2021).

19 Gao, Z. *et al.* Constructing Well-Defined and Robust Th-MOF-Supported Single-Site Copper for Production and Storage of Ammonia from Electroreduction of Nitrate. *ACS Cent. Sci.* **7**, 1066-1072 (2021).

Comment 2: *The authors must clarify the significance to reducing nitrite to ammonia, given the fact that most industrial/commercial nitrite (and even those in wastewater) were produced by oxidizing ammonia (the Ostwald process). Therefore, consuming additional energy to reduce such “nitrate” back to ammonia seems like*

to be wasted efforts (especially when using ultrapure chemicals as nitrate sources). In addition, if the authors could show that the nitrates in wastewater could be selectively turned into ammonia, this work would be much more convincing and significant. Otherwise, the better usage of collected and purified environmental nitrites is to produce nitric acid or nitrite fertilizer instead of reducing them back to ammonia.

Response: Thanks for the comment. Based on our knowledge and the massive published literature, the significance to reducing nitrate to ammonia can be concluded as follows:

- 1) Although the nitrate is currently produced by ammonia oxidation (the Ostwald process), the direct N_2 oxidation into nitrate has also been achieved by applying the advanced technologies such as the triboelectric nanogenerator (*Energy Environ. Sci.*, 2020, **13**, 2450-2458) and plasma activation (*Angew. Chem. Int. Ed.*, 2021, **60**, 14131-14137; *Joule*, 2021, 5, 3006-3030.). Given the fact that the N_2 oxidation into nitrate can be achieved with much lower energy consumption and carbon emission than its direct reduction into ammonia, great potential is expected in NO_3^- RR to accomplish the green ammonia synthesis.
- 2) The ultrahigh dissociation energy of N_2 is avoided in NO_3^- RR to accomplish more energy-favorable routes for ammonia synthesis.
- 3) The highest valence state of N-element in nitrate ensures that the deep reduction reaction can be achieved for selective ammonia synthesis. While the N_2 oxidation and reduction may proceed simultaneously when conducting catalytic N_2 RR for ammonia synthesis, which restrains the NH_4^+ selectivity.
- 4) The abundant nitrate in wastewater offers sufficient feedstocks for NO_3^- RR routes.
- 5) The wide distribution of general organic matters such as aldehydes and phenols in wastewater is noted, which forms contaminant mixture with the nitrate. That is to say, these organic matters can be functioned as what is called the hole sacrificial agents, which accelerates both the NO_3^- reduction for ammonia synthesis and pollutants oxidation for their degradation. (*related experimental results are appended below*).

The significance to reducing nitrate to ammonia is clarified and emphasized in the

revised MS file.

●**Page 3 in MS:** From an energy viewpoint, nitrate ions (NO_3^-), as a sustainable N-containing alternative, can be facilely disintegrated at a much lower dissociation energy of 204 kJ mol^{-1} , contributing to an accelerated reaction kinetics for NH_3 synthesis. Besides, the highest valence state of N-element in NO_3^- ensures that the deep reduction reaction can be achieved for selective NH_4^+ synthesis. The intermediate-valence N_2 oxidation and reduction may proceed simultaneously when conducting catalytic N_2RR for NH_4^+ synthesis, which restrains the NH_4^+ selectivity. Another virtue of using NO_3^- feedstock lies in its rich distribution in wastewater. The abundant nitrate in wastewater offers sufficient reactants for NO_3^- reduction reaction (NO_3^- RR) routes. The wide distribution of general organic matters such as aldehydes and phenols in wastewater is also noted, which forms contaminant mixture with the nitrate. These organic matters can be used as what is called the hole sacrificial agents, which accelerates both the NO_3^- reduction for NH_4^+ synthesis and pollutants oxidation for their degradation.

Most importantly, the practical application of NO_3^- RR in simulated wastewater is developed and append in the revised MS and SI. It is found that the nitrate in wastewater could be selectively turned into ammonia, which makes this work more convincing and significant.

●**Page 16 in MS:**

Practical applications of NO_3^- RR in simulated wastewater

The practical application of the as-proposed NO_3^- RR for ammonia photosynthesis route was developed. Since the ethylene glycol (EG) is applied in the catalysis system (Catal. Sys.) as the hole sacrificial agent, the conversion pathways of EG was first investigated via the *in-situ* DRIFTS technology (Fig. 6a). It is observed that dynamic adsorption equilibrium (Ads. Equil.) of EG is gradually formed based on the detection of methane ($2940, 2880, 1437$ and 1364 cm^{-1}) and alcohol ($1123, 1080$ and 1040 cm^{-1}) species. Then the generation and accumulation of formate (1153 cm^{-1}) and carbonate (1285 cm^{-1}) were observed, which were attributed to the primary products for EG oxidation. Hence, it is concluded that the reactions of EG oxidation and NO_3^-

reduction proceed simultaneously, in which the hole consumption by EG oxidation could in turn accelerate the NO_3^- RR to promote ammonia synthesis.

Most importantly, it should be noted that abundant organic contaminants are distributed in many situations of NO_3^- -containing wastewater, which can be used as what is called the hole sacrificial agent. Based on this consideration, phenol, benzyl alcohol and formaldehyde were added into the catal. sys. as potential contaminants respectively (Fig. 6b and Supplementary Fig. 50a-50f). It is found that the ammonia synthesis rates and selectivity were all retained, which indicates that the NO_3^- RR route is established in the simulated organic wastewater. Interestingly, the ammonia synthesis rates are increased in the order of conventional catal. sys. ($3.78 \text{ mmol g}^{-1} \text{ h}^{-1}$) < benzyl alcohol ($5.37 \text{ mmol g}^{-1} \text{ h}^{-1}$) < phenol ($7.64 \text{ mmol g}^{-1} \text{ h}^{-1}$) < formaldehyde ($8.99 \text{ mmol g}^{-1} \text{ h}^{-1}$). It is deduced that the hole consumption capacity of these organic contaminants is higher than that of EG, which leaves more electrons to catalyze the NO_3^- RR. Then, the corresponding test of formaldehyde was conducted without EG as the hole sacrificial agent (Fig. 6b and Supplementary Fig. 50g-50h). It is observed that the ammonia synthesis rate was further elevated to $11.14 \text{ mmol g}^{-1} \text{ h}^{-1}$ with 97.82% of selectivity noted, which illustrates that the formaldehyde can act as the “hole sacrificial agent” more efficiently than that of EG. Based on the organic contaminant investigation in the simulated wastewater, it is clarified that the NO_3^- RR route can be developed as a “sacrificial-agent-free” technology for the application of ammonia synthesis in wastewater coupled with the organic pollutants’ oxidation, which shows scientific significance in the both area of environmental remediation and energy conversion. Besides, the addition of cation contaminants (Co^{2+} , Ni^{2+} and Cd^{2+} , Supplementary Fig. 51) and anion contaminants (SO_4^{2-} , PO_4^{3-} and CO_3^{2-} , Supplementary Fig. 52) have also been considered, in which the catalytic efficiency is accomplished in general. Some discrepancy in the performance is noted among these ions, which could be raised by the complicated impact of added ions on the catal. sys. and requires further investigation in the future.

Fig. 6, Practical application of NO_3^-RR to NH_4^+ route in simulated wastewater. a, *In-situ* DRIFTS for EG (hole sacrificial agent) oxidation during the NO_3^-RR . b, Ammonia synthesis rates and selectivity evaluation by adding different types of simulated wastewater into the catalysis system (Catal. Sys.), including the organic matter (phenol, benzyl alcohol and formaldehyde), cation (Co^{2+} , Ni^{2+} and Cd^{2+}) and anion (SO_4^{2-} , PO_4^{3-} and CO_3^{2-}) contaminants correspondingly. As for the condition of formaldehyde, the catalytic tests were conducted with and without EG respectively.

●Page 33 in SI:

Supplementary Figure 50. Raw data of the efficiency evaluation in simulated wastewater containing the phenol (a, b), benzyl alcohol (c, d) and formaldehyde (e, f) as organic contaminants. The EG is excluded as hole sacrificial agent for the formaldehyde test (g, h)

Supplementary Figure 51. Raw data of the efficiency evaluation in simulated wastewater containing the Co²⁺ (a, b), Ni²⁺ (c, d) and Cd²⁺ (e, f) as cation contaminants.

Supplementary Figure 52. Raw data of the efficiency evaluation in simulated wastewater containing the SO₄²⁻ (a, b), PO₄³⁻ (c, d) and CO₃²⁻ (e, f) as anion contaminants.

Comment 3: The author claimed that the construction of MO_{NCs} and the improvement of NH₄⁺ photosynthesis rate were completed at the same time. If it is done simultaneously, the ammonia production rate (increasing slope) should show a gradual increase as the reaction time progresses, However, Fig. 3a shows that the catalyst maintains a stable ammonia production rate (constant slope). I suggest clarifying that point. Or, what's the timeline for the formation of MO_{NCs} on TNS?

Response: Thanks for the carefulness of the Reviewer. As depicted in Fig. 3a, the slope is not consistent across the whole curve (*refer the enlarged profile in Fig. R1*). Since the construction of MO_{NCs} (Fig. 1f and Supplementary Fig. 8) and NH₄⁺ synthesis proceed with the same time, the gradual increase of the slope for NH₄⁺ generation is reasonable. This point is clarified in the revised MS.

Page 9 in MS: Most importantly, the *operando* construction of MO_{NCs} (illustrated in Fig. 1f) and enhancement of the NH₄⁺ photosynthesis rate are simultaneously accomplished with the reaction on stream. **Since the construction of MO_{NCs} and NH₄⁺ synthesis proceed at the same time, the NH₄⁺ synthesis rate by MO_{NCs} is elevated and the gradual increase of the slope for NH₄⁺ generation is reasonable.**

Figure R1. Enlarged profile of the NH_4^+ synthesis rate on $\text{BaO}_{\text{NCS}}\text{-TNS}$ showing the increasing slope.

Comment 4: The authors may need to explain the y-axis for Fig. 3c,d.

Response: Thanks for the suggestion. The y-axis for Fig. 3c and 3d is explained and added in the revised MS.

•**Page 11 in MS: c, d** Quantitative isotope-labelled $^{15}\text{NO}_3^-$ study verifying the fed NO_3^- as the source for the produced NH_3 . Inset: raw IC spectra for $^{14}\text{NO}_3^-/^{15}\text{NO}_3^-$ reduction (c) and $^{14}\text{NH}_4^+/^{15}\text{NH}_4^+$ generation (d) respectively. The y-axis of Fig. 3c and 3d depict the reaction rates for NO_3^- reduction and NH_4^+ production respectively.

Comment 5: In the photocatalytic reaction operation method described by the author (Lines 415–430), clusters are grown on TNS in the catalytic reaction. Would the catalyst need to be washed? Nevertheless, the catalyst was washed in the catalyst synthesis (Line 356). This part needs further clarity.

Response: Thanks for the comments. This confusing description is updated in the revised MS file.

•**Page 21 in MS:** After photocatalysis reaction, the *operando* construction of MO_{NCS} on TNS is accomplished. The photocatalysts were collected and washed for further characterization.

Comment 6: As shown in Fig 3a, TNS itself exhibited nice catalytic activity. The

active sites in TNS and MO_{NCs}-TNS should be elucidated more clearly in the manuscript.

Response: Thanks for the helpful suggestion. The active sites in TNS are identified as the oxygen vacancies (OVs) due to the observable OVs construction via light irradiation (Fig. 2b and Supplementary Fig. 11). Besides, the *operando* production of MO_{NCs} on TNS is preferable to achieve at the OVs sites (Fig. 2c and Supplementary Figs. 12-15). Therefore, the active sites in MO_{NCs}-TNS should be the MO_{NCs}@OVs interfaces. The corresponding description is added in the revised MS.

●**Page 9 in MS:** As depicted in Fig. 3a, the pristine TNS exhibited nice catalytic activity (1.65 mmol g_{cat}⁻¹ h⁻¹). The active sites in TNS are identified as the oxygen vacancies (OVs) due to the observable OVs construction via light irradiation (Fig. 2b and Supplementary Fig. 12).

●**Page 9 in MS:** Since the *operando* production of MO_{NCs} on TNS is preferable to achieve at the OVs sites (Fig. 2c and Supplementary Figs. 12-15), the active sites in MO_{NCs}-TNS is regarded as the MO_{NCs}@OVs interfaces.

Comment 7: In Fig 4c, the energy of the transition state is not marked. Then, how to calculate the energy difference of 1.42 eV?

Response: Thanks for the comment. The energy barriers were calculated to determine the transition states (Supplementary Fig. 36). The energy value is marked into the revised Fig. 4.

●**Page 12 in MS:**

Fig. 4 Selectivity and long-term stability tests. **a**, NH_4^+ selectivity test from NO_3^- RR versus the other potential products. The related stand curves were provided in Supplementary Figs. 37-39. **b**, Comparison of NH_4^+ production rate and selectivity with different ammonia synthesis routes under ambient conditions. **c**, Calculated activation energy for NO_3^- RR for NH_4^+ synthesis and water splitting for H_2 generation. **d**, Long-term stability of $\text{BaO}_{\text{NCS}}\text{-TNS}$ and comparison of total NH_4^+ yield with different NH_4^+ synthesis routes.

•Page 23 in SI:

Supplementary Figure 36. Calculated reaction coordinates of NO_3^- reduction (**a**) and water splitting (**b**). The calculations were considered as relaxation when the single imaginary frequency

(f/i) was located. The blue, red, green, light grey and pink spheres depict Ti, O, Ba, N and H atoms, respectively.

Comment 8: *The author needs to calculate band gaps and band edges through Mott-Schottky, XPS valence band spectrum, diffuse reflectance spectrum, etc.*

Response: Thanks for the helpful suggestion. The related characterization including Mott-Schottky spectra, XPS VB spectra and UV-vis DRS are appended in the revised MS and SI (Supplementary Fig. 20). The corresponding description is also added.

●**Page 8 in MS:** Then the Mott-Schottky spectra and UV-vis DRS were combined to determine the band structures of TNS and BaO_{NCS}-TNS (Supplementary Fig. 19). It is noted that the conduction band position of BaO_{NCS}-TNS is elevated than that of the pristine TNS, which enhances the reduction capacity for the elevated NO₃⁻RR performance.

●**Page 20 in MS:** The Mott-Schottky spectra were conducted using Pt/C, Ag/AgCl and Pt as working, reference and counter electrodes respectively on an electrochemical workstation (CHI-660E), and the results were recorded from -1.0 V to 1.5 V at 1000 Hz without light irradiation.

●**Page 14 in SI:** Since the contaminative C element is inevitable during the XPS tests, the band edge cannot be precisely characterized by the XPS VB spectra. Hence the Mott-Schottky spectra and UV-vis DRS were combined to determine the band structures of TNS and BaO_{NCS}-TNS.

Supplementary Figure 19. Estimated band structures. Mott-Schottky spectra (a), estimated band gap calculated by UV-vis DRS (b), XPS VB spectra (c) and illustration for the band structures of TNS and BaO_{NCs} -TNS.

Comment 9: *Further characterization like XAFS of Ba_{SAs} and BaO_{NCs} should be supplemented to unambiguously describe a more realistic structure of active sites.*

Response: Thanks for the suggestion. After evaluating the feasibility for conducting the corresponding XAFS tests with some XAFS experts and operators, we have been informed that the EXAFS signals of Ba element will be distracted by that of the TiO_2 substrate, which makes us unable to identify the Ba single atoms or BaO clusters on the TNS surface. We apologize that the XAFS results cannot be provided in this research system. This technological problem is not solvable unless the Ba species are replaced by some transitional or noble elements. However, by combining the HAADF-STEM images (Fig.1a-1d), IC detection for Ba^{2+} concentration (Fig. 1f), XPS (Fig. 2a) and DFT calculations (Fig. 1c), we still believe that the identification of BaO_{NCs} is convincing.

Comment 10: In Supplementary Figs. 5 and 6, the elemental distribution of Mg and Ca (esp. Ca) were not overlapped with TNS. Why? Do these isolated MO clusters exhibit catalytic activity? The authors may need to clarify this point.

Response: Thanks for this helpful comment. The bright spots of elemental mapping images do not indicate that the corresponding element is located at this exact spot. It just shows that some elements were detected in the neighboring area. Besides, some inevitable error is generally noted in the elemental mapping testing procedure, such as drift of test samples, impact of the additional signals from the substrate and the impurities in the TEM tube. The comprehensive results are presented in one mapping figure. Thus, it is hard to precisely locate some elements at specific sites by this technology. However, in the HAADF-STEM results (Fig. 1a-1d and Supplementary Fig. 4), the MO_{NCs} is well-overlapped with TNS. It is reasonable that the superior ammonia synthesis efficiency is contributed by the construction of MO_{NCs} -TNS composites rather than the pristine MO_{NCs} .

In order to further unveil the activity origin of MO_{NCs} -TNS, we conduct additional control experiment by replacing the TNS substrate with the photocatalytic inert SiO_2 nanoparticles. It is observed that the no NH_4^+ is detected during the simultaneous construction of MO_{NCs} - SiO_2 and NO_3^- RR. Hence, it is confirmed that these individual MO clusters exhibit no catalytic activity. The corresponding results are appended in the revised MS and SI files.

●**Page 9 in MS:** Besides, in order to further unveil the activity origin of MO_{NCs} -TNS, we conduct additional control experiment by replacing the TNS substrate with the photocatalytic inert SiO_2 nanoparticles (Supplementary Fig. 24). It is observed that the no NH_4^+ is detected during the simultaneous construction of MO_{NCs} - SiO_2 and NO_3^- RR. Hence, it is clarified that these individual MO clusters exhibit no catalytic activity, in which the superior NH_4^+ synthesis efficiency gives credit to the construction of MO_{NCs} -TNS composites.

●**Page 16 in SI:**

Supplementary Figure 24. Control experiment towards NO₃⁻RR to NH₄⁺ synthesis by replacing TiO₂ with SiO₂. The test parameters were the same with that of Fig. 3a.

Comment 11: The direct comparison in selectivity/activity for NO₃⁻RR and N₂RR is unfair and unnecessary. The scientific significance and the difficulty for the activation of N-N triple bond (nitrogen fixation) is much higher than just reducing nitrite (most of which comes from ammonia) back to ammonia.

Response: Thanks for the helpful suggestion. The comparison between this manuscript and reported works is revised and enriched, especially appending the catalytic efficiency comparison with other photocatalytic/electrocatalytic NO₃⁻RR work (Fig. 4, refer Supplementary Table 1-2 for the complete lists). It is evidently observed that the superior systematic ammonia synthesis efficiency is accomplished, exceeding that of the other photocatalytic NO₃⁻RR works and even some leading electrocatalytic NO₃⁻RR work (*Nat. Energy*, 2020, 5, 605-613). Not surprisingly, the results of this work also exhibit significant advances in comparison with that of these N₂RR routes at ambient condition, including electrocatalytic, photocatalytic and photoelectrochemical methods.

Page 2 in Abstract: The catalyst exhibits an ultrahigh ammonia photosynthesis rate of 11.97 mol g_{metal}⁻¹ h⁻¹ (89.79 mmol g_{cat}⁻¹ h⁻¹) with nearly 100% selectivity. A total ammonia yield of 0.78 mmol within 72 hours is achieved, which exhibits significant advantage in the area of photocatalytic NO₃⁻ and even exceeds some of the leading electrocatalytic NO₃⁻RR work.

●**Page 4 in MS:** As a result, the BaO_{NCs}-TNS composite achieves an ultrahigh rate for sustainable NH₄⁺ photosynthesis (11.97 mol g_{Ba}⁻¹ h⁻¹ or 89.79 mmol g_{cat}⁻¹ h⁻¹), which achieves a milestone value for the photocatalytic NO₃⁻RR routes and exhibits significant advantage in the other ammonia synthesis routes under ambient condition.

●**Page 12 in MS:** The NH₄⁺ synthesis rate and selectivity of the NO₃⁻RR on BaO_{NCs}-TNS are superior among the routes under ambient condition, exceeding that of the other photocatalytic NO₃⁻RR works and even some leading electrocatalytic NO₃⁻RR work (Fig. 4b and Supplementary Table 2). Not surprisingly, the results of this work also exhibit significant advances in comparison with that of these N₂RR routes at ambient condition, including electrocatalytic, photocatalytic and photoelectrochemical methods.

Fig. 4 Selectivity and long-term stability tests. **a**, NH₄⁺ selectivity test from NO₃⁻RR versus the other potential products. The related stand curves were provided in Supplementary Figs. 37-39. **b**, Comparison of NH₄⁺ production rate and selectivity with different ammonia synthesis routes under ambient conditions. **c**, Calculated activation energy for NO₃⁻RR for NH₄⁺ synthesis and water splitting for H₂ generation. **d**, Long-term stability of BaO_{NCs}-TNS and comparison of total

NH_4^+ yield with different NH_4^+ synthesis routes.

Response to Referee 2:

Recommendation: Rejection.

General Comment: The manuscript describes that the BaOx nanoclusters loaded on TiO₂ nanosheets promote photocatalytic NO₃⁻ to NH₄⁺ reduction in water with sacrificial electron donor under UV irradiation. I cannot recommend the manuscript for publication. Followings should be considered.

Response: Thanks for the comments. To address the concern of the Reviewer, the following comments are responded point by point, and some essential evidence and analysis are appended in the revised manuscript (MS) and Supplementary Information (SI) files to make this work more significant and convincing.

At first, the ammonia synthesis efficiency was tested by using UV light (89.79 mmol g_{cat}⁻¹ h⁻¹), full spectrum (15.80 mmol g_{cat}⁻¹ h⁻¹) and simulated solar light (3.07 mmol g_{cat}⁻¹ h⁻¹) respectively in this work. Observable activity was noted under all the above conditions. We believe that the introduction of UV light is necessary to fully optimize the reaction parameters to establish the optimal reaction condition for NO₃⁻RR (Fig. 3b and Supplementary Figs. 25, 26 and 28). It is stated in the MS that the stable NO₃⁻ reactant can be preactivated by UV light to accelerate its reduction reaction, as suggested by the published work (*Environ. Sci. Technol.*, 2019, 53, 316-324.). Most importantly, in order to clarify the contribution of UV light to the NO₃⁻ preactivation, the corresponding *in-situ* DRIFTS experiment is designed and conducted, which is appended in the revised MS and SI. It is evidently found that NO₃⁻ could be preactivated by UV light, which tailors the coordination environment of the stable NO₃⁻ and drives it into some active intermediates such as monodentate NO₃⁻, -NO₂ and NO₂⁻. Thus, it is worth mentioning that **the introduction of UV light not only increase the energy density, but realize the preactivation of NO₃⁻**, which exceeds that of the full-spectrum and simulated solar light.

●**Page 12 in MS:** Besides, since NO₃⁻ could be preactivated by UV light, which tailors the coordination environment of the stable NO₃⁻ and drives it into some active intermediates such as monodentate NO₃⁻, -NO₂ and NO₂⁻ (Supplementary Fig. 27), the utilization of different light sources was tested. Notably, the optimal NH₄⁺

photosynthesis rate is $38.00 \text{ mmol g}_{\text{cat}}^{-1} \text{ h}^{-1}$ with pristine TNS after regulating the reaction parameters (Supplementary Fig. 28). Moreover, an ultrahigh rate for NH_4^+ photosynthesis is accomplished on $\text{BaO}_{\text{NCs}}\text{-TNS}$ ($89.79 \text{ mmol g}_{\text{cat}}^{-1} \text{ h}^{-1}$) under the same reaction conditions with UV light. It is worth mentioning that the introduction of UV light not only increase the energy density, but realize the preactivation of NO_3^- , which exceeds that of the full-spectrum ($15.80 \text{ mmol g}_{\text{cat}}^{-1} \text{ h}^{-1}$) and simulated solar light ($3.07 \text{ mmol g}_{\text{cat}}^{-1} \text{ h}^{-1}$).

●Page 18 in SI:

Supplementary Figure 27. *In-situ* DRIFTS results for the preactivation of NO_3^- by UV light irradiation.

The test procedure is similar with that of the *in-situ* DRIFTS test in Fig. 5b by replacing the NO_3^- -contained catalyst with pristine KNO_3 powder. It is observed in the above figure that some bridging NO_3^- (1631 and 1173 cm^{-1})^{1,2} is transferred into the monodentate NO_3^- species (1513 , 1283 , 1070 and 1049 cm^{-1})^{2,3}, which is more active than that of the bridging NO_3^- ⁴. Moreover, the intermediated nitro ($-\text{NO}_2$, 1432 and 1374 cm^{-1}) and NO_2^- (824 cm^{-1}) are obviously increased.⁴ It is thus concluded that the NO_3^- reactant can be preactivated by UV light irradiation.

Supplementary reference:

1 Zhang, X. L., He, H., Cao, H. W. & Yu, Y. B. Experimental and theoretical studies of surface nitrate species on $\text{Ag}/\text{Al}_2\text{O}_3$ using DRIFTS and DFT. *Spectrochim. Acta A*, **71**,

1446-1451 (2008).

2 Yang, X. F. *et al.* DRIFTS Study of Ammonia Activation over CaO and Sulfated CaO for NO Reduction by NH₃. *Environ. Sci. Technol.*, **45**, 1147-1151 (2011).

3 Cui, W. *et al.* Highly Efficient Performance and Conversion Pathway of Photocatalytic NO Oxidation on SrO-Clusters@Amorphous Carbon Nitride. *Environ. Sci. Technol.*, **51**, 10682-10690 (2017).

4 Hadjiivanov, K. Identification of Neutral and Charged N_xO_y Surface Species by IR Spectroscopy. *Catal. Rev. Sci. Eng.*, **42**, 71-144 (2000).

As for the sacrificial electron donor issue, the investigation of practical application of NO₃⁻RR in simulated wastewater is appended. It is found that the nitrate in wastewater could be selectively turned into ammonia, which makes this work more solid. Most importantly, the “sacrificial-agent-free” route is established by replacing EG (sacrificial electron donor) with the formaldehyde contaminant, which shows scientific significance in both reducing energy consumption and mitigating environmental anxieties.

●**Page 16 in MS:**

Practical applications of NO₃⁻RR in simulated wastewater

The practical application of the as-proposed NO₃⁻RR for ammonia photosynthesis route was developed. Since the ethylene glycol (EG) is applied in the catalysis system (Catal. Sys.) as the hole sacrificial agent, the conversion pathways of EG was first investigated via the *in-situ* DRIFTS technology (Fig. 6a). It is observed that dynamic adsorption equilibrium (Ads. Equil.) of EG is gradually formed based on the detection of methane (2940, 2880, 1437 and 1364 cm⁻¹) and alcohol (1123, 1080 and 1040 cm⁻¹) species. Then the generation and accumulation of formate (1153 cm⁻¹) and carbonate (1285 cm⁻¹) were observed, which were attributed to the primary products for EG oxidation. Hence, it is concluded that the reactions of EG oxidation and NO₃⁻ reduction proceed simultaneously, in which the hole consumption by EG oxidation could in turn accelerate the NO₃⁻RR to promote ammonia synthesis.

Most importantly, it should be noted that abundant organic contaminants are distributed in many situations of NO₃⁻-containing wastewater, which can be used as

what is called the hole sacrificial agent. Based on this consideration, phenol, benzyl alcohol and formaldehyde were added into the catal. sys. as potential contaminants respectively (Fig. 6b and Supplementary Fig. 50a-50f). It is found that the ammonia synthesis rates and selectivity were all retained, which indicates that the NO_3^- RR route is established in the simulated organic wastewater. Interestingly, the ammonia synthesis rates are increased in the order of conventional catal. sys. ($3.78 \text{ mmol g}^{-1} \text{ h}^{-1}$) < benzyl alcohol ($5.37 \text{ mmol g}^{-1} \text{ h}^{-1}$) < phenol ($7.64 \text{ mmol g}^{-1} \text{ h}^{-1}$) < formaldehyde ($8.99 \text{ mmol g}^{-1} \text{ h}^{-1}$). It is deduced that the hole consumption capacity of these organic contaminants is higher than that of EG, which leaves more electrons to catalyze the NO_3^- RR. Then, the corresponding test of formaldehyde was conducted without EG as the hole sacrificial agent (Fig. 6b and Supplementary Fig. 50g-50h). It is observed that the ammonia synthesis rate was further elevated to $11.14 \text{ mmol g}^{-1} \text{ h}^{-1}$ with 97.82% of selectivity noted, which illustrates that the formaldehyde can act as the “hole sacrificial agent” more efficiently than that of EG. Based on the organic contaminant investigation in the simulated wastewater, it is clarified that the NO_3^- RR route can be developed as a “sacrificial-agent-free” technology for the application of ammonia synthesis in wastewater coupled with the organic pollutants’ oxidation, which shows scientific significance in the both area of environmental remediation and energy conversion. Besides, the addition of cation contaminants (Co^{2+} , Ni^{2+} and Cd^{2+} , Supplementary Fig. 51) and anion contaminants (SO_4^{2-} , PO_4^{3-} and CO_3^{2-} , Supplementary Fig. 52) have also been considered, in which the catalytic efficiency is accomplished in general. Some discrepancy in the performance is noted among these ions, which could be raised by the complicated impact of added ions on the catal. sys. and requires further investigation in the future.

Fig. 6, Practical application of NO_3^- RR to NH_4^+ route in simulated wastewater. a, *In-situ* DRIFTS for EG (hole sacrificial agent) oxidation during the NO_3^- RR. b, Ammonia synthesis rates and selectivity evaluation by adding different types of simulated wastewater into the catalysis system (Catal. Sys.), including the organic matter (phenol, benzyl alcohol and formaldehyde), cation (Co^{2+} , Ni^{2+} and Cd^{2+}) and anion (SO_4^{2-} , PO_4^{3-} and CO_3^{2-}) contaminants correspondingly. As for the condition of formaldehyde, the catalytic tests were conducted with and without EG respectively.

●Page 33 in SI:

Supplementary Figure 50. Raw data of the efficiency evaluation in simulated wastewater containing the phenol (a, b), benzyl alcohol (c, d) and formaldehyde (e, f) as organic contaminants. The EG is excluded as hole sacrificial agent for the formaldehyde test (g, h)

Supplementary Figure 51. Raw data of the efficiency evaluation in simulated wastewater containing the Co²⁺ (a, b), Ni²⁺ (c, d) and Cd²⁺ (e, f) as cation contaminants.

Supplementary Figure 52. Raw data of the efficiency evaluation in simulated wastewater containing the SO_4^{2-} (a, b), PO_4^{3-} (c, d) and CO_3^{2-} (e, f) as anion contaminants.

*Comment 1: Almost 100% selectivity for NO_3^- to NH_4^+ on bare TiO_2 has already been reported (Hirakawa, H.; Hashimoto, M.; Shiraishi, Y.; Hirai, T. *Selective Nitrate-to-Ammonia Transformation on Surface Defects of Titanium Dioxide Photocatalysts*. *ACS Catal.* 2017, 7 (5), 3713–3720.). This report was not cited in the manuscript. The results presented in the present manuscript are therefore not surprising.*

Response: Thanks for the comment. We apologize for the carelessness conducting literature survey. The provided reference is cited in the revised MS. It is true that the almost 100% selectivity for NO_3^- to NH_4^+ has been accomplished by some reports. We still believe that this manuscript shows significance on many indicators evaluating catalytic performance including ammonia synthesis efficiency, selectivity, long-term stability and total ammonia yield. This conclusion should be credible due to the application of systematic activity evaluation procedure and comprehensive detection for potential products in this manuscript. The comparison between this manuscript and reported works is revised and enriched (Fig. 4, refer Supplementary Table 1-2 for the complete lists). It is evidently observed that the superior systematic ammonia synthesis efficiency is accomplished, exceeding that of the other photocatalytic NO_3^- RR works and even some leading electrocatalytic NO_3^- RR work (*Nat. Energy*, 2020, 5, 605-613). Not surprisingly, the results of this work also exhibit advances in comparison with those of N_2 RR routes at ambient condition, including electrocatalytic, photocatalytic and photoelectrochemical methods.

•Page 11 in MS: By comparing the eight-electron NO_3^- RR and two-electron water splitting reaction, the selectivity for NH_4^+ photosynthesis is determined to be as high as 97.67%. The NH_4^+ synthesis rate and selectivity of the NO_3^- RR on BaO_{NCs} -TNS are superior among the routes under ambient condition, exceeding that of the other photocatalytic NO_3^- RR works and even some leading electrocatalytic NO_3^- RR work (Fig. 4b and Supplementary Table 2). Not surprisingly, the results of this work also

exhibit advances in comparison with those of N_2 RR routes at ambient condition, including electrocatalytic, photocatalytic and photoelectrochemical methods.

•Page 24 in MS:

Ref. 16: Hirakawa, H., Hashimoto, M., Shiraishi, Y. & Hirai, T. Selective Nitrate-to-Ammonia Transformation on Surface Defects of Titanium Dioxide Photocatalysts. *ACS Catal.* **7**, 3713-3720 (2017).

Fig. 4 Selectivity and long-term stability tests. **a**, NH_4^+ selectivity test from NO_3^- RR versus the other potential products. The related stand curves were provided in Supplementary Figs. 37-39. **b**, Comparison of NH_4^+ production rate and selectivity with different ammonia synthesis routes under ambient conditions. **c**, Calculated activation energy for NO_3^- RR for NH_4^+ synthesis and water splitting for H_2 generation. **d**, Long-term stability of BaONCS-TNS and comparison of total NH_4^+ yield with different NH_4^+ synthesis routes.

Other than record high catalytic performance of NO_3^- RR for ammonia photosynthesis in this manuscript, the following significance and novelty should also be noted:

1) A general strategy is built to facilyly construct the subnanometric clusters without

ligands and harsh condition, which can facilitate the development of cluster catalysis.

2) Comprehensive reaction mechanism is proposed by combined *in-situ* technology and DFT calculations, which promote the establishment of the analytical methods at molecular-level, which helps to investigate the nature of general catalysis reactions.

3) The practical application of NO₃⁻RR in simulated wastewater is appended in the revised MS and SI, which provides a feasible and novel approach to reduce energy consumption and mitigate environmental anxieties.

Comment 2: The authors claimed that the BaO_x nanoclusters behave as active sites for NO₃⁻ reduction only by DFT calculations. This is very weak to support the assumptions provided here. In the present system, many Cl⁻ ions present in the solution (see the experimental section). The possible mechanism for the reaction enhancement by the addition of BaCl₂ to the solution is that Cl⁻ behaves as a sacrificial electron donor (see: Huang, L.; Li, R.; Chong, R.; Liu, G.; Han, J.; Li, C. Cl Making Overall Water Splitting Possible on TiO₂-Based Photocatalysts. Catal. Sci. Technol. 2014, 4 (9), 2913–2918).

Based on the above, I do not think the novelty and quality of this manuscript justifies publication in Nat. Commun.

Response: Thanks for the comments. The active sites of MO_{NCs} were supported by combined catalyst characterization, efficiency test and DFT calculations in the previous version. In order to fully unveil the origin of enhanced activity, some experiments are appended in the revised MS and SI files (listed below in blue). By providing these comprehensive evidence (listed below 1-8), we believe that the in-depth understanding of the MO_{NCs} can be established, including their morphology, chemical structures, formation mechanism, charge transfer properties and contribution to the NO₃⁻RR efficiency. Then the potential contribution of Cl⁻ to the ammonia synthesis efficiency is fully addressed below.

1) HAADF-STEM and related elemental mappings reveal the *operando* generation of MO_{NCs} (Fig. 1a-1e, Supplementary Figs. 4-7).

2) IC detection for the M²⁺ decrease in the reaction solution illustrates the growth of

MO_{NCs} on TNS surface (Fig. 1f and Supplementary Fig. 8).

3) XPS and XRF analysis for Ba element identify its content, valence state and chemical composition (Fig. 2a, Supplementary Figs. 9-10).

4) Combined EPR tests and formation energy calculations clarify that the deposition and growth of MO_{NCs} tend to proceed at the OV sites of TNS (Fig. 2b-2c, Supplementary Figs. 11-15), which provides the formation mechanism for MO_{NCs}-TNS composites.

5) Steady and time-resolved PL spectra (Fig. 2d and Supplementary Figs. 16-17) unravels the elevated charge transfer capacity by MO_{NCs} construction.

6) Appending combined UV-vis DRS, XPS-VB spectra and Mott-Schottky results (Supplementary Fig. 19) prove that the photocatalytic reduction efficiency is enhanced by MO_{NCs} construction, which accelerates the NO₃⁻RR.

●**Page 8 in MS:** Then the Mott-Schottky spectra and UV-vis DRS were combined to determine the band structures of TNS and BaO_{NCs}-TNS (Supplementary Fig. 19). It is noted that the conduction band position of BaO_{NCs}-TNS is elevated than that of the pristine TNS, which enhances the reduction capacity for the elevated NO₃⁻RR performance.

●**Page 13 in SI:**

Supplementary Figure 19. Estimated band structures. Mott-Schottky spectra (a), estimated band gap calculated by UV-vis DRS (b), XPS VB spectra (c) and illustration for the band structures of

TNS and BaO_{NCs}-TNS.

Since the contaminative C element is inevitable during the XPS tests, the band edge cannot be precisely characterized by the XPS VB spectra. Hence the Mott-Schottky spectra and UV-vis DRS were combined to determine the band structures of TNS and BaO_{NCs}-TNS.

7) Appending the PBE+*U* correction for potential energy calculations (Fig. 2e and Supplementary 20-22) further specifies the electron migration direction at the MO_{NCs}/TNS interfaces.

●Page 8 in MS: Molecular-level insights into the charge transfer patterns at MO_{NCs}/TNS interface were further revealed by DFT calculations (Fig. 2e and Supplementary Figs. 20-22). As supported by both standard PBE functional and PBE+*U* correction calculations for the planar average potential energy profile, a distinct energy gap is generated between the BaO_{NCs} and TNS surface, which facilitates directional electron migration from the BaO_{NCs} to TNS.

Figure 2e. Calculated planar average potential energy profile using respective PBE functional and PBE+*U* correction. Inset: calculated charge difference distribution at the BaO_{NCs}/TNS interface, in which charge accumulation is marked in blue and charge depletion is marked in yellow. The isosurface was set to 0.005 eV Å⁻³.

8) Appending controlled activity test unveils that the activity origin is contributed by the construction of MO_{NCs}/TNS interfaces.

●Page 9 in MS: Besides, in order to further unveil the activity origin of MO_{NCs}-TNS, we conduct additional control experiment by replacing the TNS substrate with the

photocatalytic inert SiO₂ nanoparticles (Supplementary Fig. 24). It is observed that the no NH₄⁺ is detected during the simultaneous construction of MO_{NCs}-SiO₂ and NO₃⁻RR. Hence, it is clarified that these individual MO clusters exhibit no catalytic activity, in which the superior NH₄⁺ synthesis efficiency gives credit to the construction of MO_{NCs}-TNS composites.

Supplementary Figure 24. Control experiment towards NO₃⁻RR to NH₄⁺ synthesis by replacing TiO₂ with SiO₂. The test parameters were the same with that of Fig. 3a.

As for the impact of Cl⁻ on the photocatalytic activity, it is true that the Cl⁻ was introduced in this catalysis system since the MCl₂•xH₂O was used as the source of alkaline earth cations. After reading the given reference (Huang, L.; Li, R.; Chong, R.; Liu, G.; Han, J.; Li, C. Cl⁻ Making Overall Water Splitting Possible on TiO₂-Based Photocatalysts. *Catal. Sci. Technol.* 2014, 4 (9), 2913–2918), it is concluded that the contribution of Cl⁻ owed to its involvement with the intermediate of O₂ evolution from water oxidation, which is not consistent with our routes in this manuscript. Most importantly, controlled experiment is conducted by adding KCl (50 and 200 mg L⁻¹ of Cl⁻ respectively) into the catalysis system of pristine TNS without other cations or anions (Supplementary Fig. 23). It is observed that the NO₃⁻RR to ammonia efficiency is not promoted by the potential involvement of Cl⁻. The related description is appended in the revised MS and SI files to identify that the enhanced activity is contributed by the construction of MO_{NCs}-TNS.

•Page 9 in MS: In addition, controlled experiment is conducted by adding KCl into the catalysis system of pristine TNS without other cations or anions (Supplementary

Fig. 23), which helps to investigate the potential involvement of Cl^- from the source of alkaline earth source ($\text{MCl}_2 \cdot x\text{H}_2\text{O}$). It is observed that the NO_3^- RR to ammonia efficiency is not promoted by the addition of Cl^- , which identifies that the enhanced activity is contributed by the construction of $\text{MO}_{\text{NCs}}\text{-TNS}$.

●Page 16 in SI:

Supplementary Figure 23. Controlled experiment by adding Cl^- into the catalysis system of pristine without other cations or anions.

Response to Referee 3:

Recommendation: Publish after minor revisions.

General Comment: A highly active and selective nitrate to NH_4^+ photocatalysis was achieved by operando construction of subnanometric alkaline-earth oxide clusters on TiO_2 nanosheets. The optimal ammonia synthesis rate and total ammonia synthesis yield are promising compared to other reported systems (Table S1). The authors support their findings with a plethora of experimental and computational techniques (XRD, SEM, TEM, STEM, elemental mapping, XPS, IC, XRF, EPR, DFT calculations, UV-VIS, FTIR, NMR, ESR, XRF, isotope labeled studied). The analysis and figures are of high technical quality. The manuscript is quite well-written. The supporting information contains many useful details, including blank experiments. This work should be of significance to the field and related fields and appears original. The work supports the conclusions and claims. The methodology seems sound and meets the expected standards in the field. The work seems publishable with minor revisions.

Response: Thanks for helpful comments and suggestions. The following comments are fully addressed point by point and the related revisions are made in the manuscript (MS) and Supplementary information (SI) files.

Comment 1: DFT calculations using the standard PBE functional predicted electron migration from the BaO_{NCs} to TNS. If the more accurate PBE+U or a hybrid functional is used, does the direction of electron migration remain the same? In systems with d- and f- localized electrons the overestimation of electron delocalization is a known weakness of PBE

Response: Thanks for the suggestion. The PBE+U correction is appended to make the calculated PBE results more convincing (Table R1). It is observed that the direction of electron migration at the MO_{NCs} -TNS interfaces remain the same. The corresponding revision is made in the MS and SI files, and the description of calculation methods is updated.

•Page 8 in MS: As supported by both standard PBE functional and PBE+U

correction calculations for the planar average potential energy profile, a distinct energy gap is generated between the BaO_{NCs} and TNS surface, which facilitates directional electron migration from the BaO_{NCs} to TNS.

●**Page 20 in MS:** The spin-polarized DFT calculations were operated with the “Vienna *ab initio* simulation package” (VASP 5.4), in which the Perdew-Burke-Ernzerhof (PBE) exchange correlation functional was included. The PBE+*U* correction (*U* = 4.0 eV) was implemented to account for the on-site charge interaction of the d electrons in Ti elements, which improved the accuracy for the calculations of electron migration at the MO_{NCs}-TNS interfaces.

●**Page 24 in MS:**

Ref. 54: Dudarev, S. L., Botton, G. A., Savrasov, S. Y., Humphreys, C. J. & Sutton, A. P. Electron-Energy-Loss Spectra and the Structural Stability of Nickel Oxide. *Phys. Rev. B: Condens. Matter Mater. Phys.* **57**, 1505-1509 (1998).

Table R1 Calculated total energy by PBE functional with +*U* correction.

Configuration	E_{tot} (PBE), eV	E_{tot} (PBE+ U), eV
MgO _{NCs} -TNS	-1720.45	-1504.70
CaO _{NCs} -TNS	-1762.14	-1597.43
SrO _{NCs} -TNS	-1738.51	-1523.35
BaO _{NCs} -TNS	-1758.98	-1542.77

Fig. 2 Chemical composition and electronic structure. **a**, Ba $3d$ XPS spectra of $\text{BaO}_{\text{NCs}}\text{-TNS}$. **b**, Room temperature solid EPR results of TNS before and after irradiation for 30 min. **c**, Calculated binding energy of BaO_{NCs} deposited at pristine and deficient TNS surfaces respectively. **d**, Time-resolved fluorescence emission decay spectra. Inset: UV-vis DRS results. **e**, Calculated planer average potential energy profile using respective PBE functional and PBE+ U correction. Inset: calculated charge difference distribution at the $\text{BaO}_{\text{NCs}}/\text{TNS}$ interface, in which charge accumulation is marked in blue and charge depletion is marked in yellow. The isosurface was set to $0.005 \text{ eV \AA}^{-3}$.

Supplementary Figure 20. Calculated charge difference density (a) and planer average potential energy profile (b) for the interface between MgO_{NCs} and TNS. The charge accumulation is labeled in blue and charge depletion is labeled in yellow. The isosurface was set to $0.0047 \text{ eV \AA}^{-3}$. Blue, red and orange spheres depict Ti, O and Mg atoms, respectively.

Supplementary Figure 21. Calculated charge difference density (a) and planer average potential energy profile (b) for the interface between CaO_{NCs} and TNS. The charge accumulation is labeled in blue and charge depletion is labeled in yellow. The isosurface was set to $0.005 \text{ eV \AA}^{-3}$. Blue, red and cyan spheres depict Ti, O and Ca atoms, respectively.

Supplementary Figure 22. Calculated charge difference density (a) and planer average potential energy profile (b) for the interface between SrO_{NCs} and TNS. The charge accumulation is labeled in blue and charge depletion is labeled in yellow. The isosurface was set to 0.005 eV Å⁻³. Blue, red and purple spheres depict Ti, O and Sr atoms respectively.

Comment 2: What was the pH of the system? I did not see it stated. Does the pH of the system change with time?

Response: Thanks for the comment. The pH value of the system during catalytic test is appended in the revised MS and SI.

●**Page 21 in MS:** The pH of this catalytic system remains at ca. 7.0 during the test since the KNO₃ and EG consist of neutral solutions (Supplementary Fig. 53).

●**Page 35 in SI:**

Supplementary Figure 53. pH value variation during the catalytic test.

Comment 3: Were all measurements done at room temperature? I did not see this clearly stated.

Response: Thanks for the comment. The temperature of the system during catalytic test is appended in the revised MS and SI.

●**Page 21 in MS:** The temperature is controlled at 25°C by using the circulating chiller (Supplementary Fig. 54).

●**Page 35 in SI:**

Supplementary Figure 54. Temperature variation during the catalytic test.

Comment 4: Abstract: The authors wrote: “A total ammonia yield of 0.78 mmol within 72 hours is achieved, which is even superior to some electro-driven N₂ to ammonia synthesis routes under ambient conditions.” This does not seem to be a fair comparison, as electro-driven N₂ to ammonia activity under ambient conditions is very low and a different reaction. It would be a fairer comparison to contrast against electro-driven nitrate to ammonia synthesis routes.

Response: Thanks for the helpful suggestion. The comparison between this manuscript and reported works is revised and enriched, especially appending the catalytic efficiency comparison with other photocatalytic/electrocatalytic NO₃⁻RR work (Fig. 4, refer Supplementary Table 1-2 for the complete lists). It is evidently observed that the superior systematic ammonia synthesis efficiency is accomplished,

exceeding that of the other photocatalytic NO_3^- RR work. Not surprisingly, the results of this work also exhibit advances in comparison with those of N_2 RR routes at ambient condition, including electrocatalytic, photocatalytic and photoelectrochemical methods.

●**Page 2 in Abstract:** The catalyst exhibits an ultrahigh ammonia photosynthesis rate of $11.97 \text{ mol g}_{\text{metal}}^{-1} \text{ h}^{-1}$ ($89.79 \text{ mmol g}_{\text{cat}}^{-1} \text{ h}^{-1}$) with nearly 100% selectivity. A total ammonia yield of 0.78 mmol within 72 hours is achieved, which exhibits significant advantage in the area of photocatalytic NO_3^- and even exceeds some of the leading electrocatalytic NO_3^- RR work

●**Page 4 in MS:** As a result, the BaO_{NCs} -TNS composite achieves an ultrahigh rate for sustainable NH_4^+ photosynthesis ($11.97 \text{ mol g}_{\text{Ba}}^{-1} \text{ h}^{-1}$ or $89.79 \text{ mmol g}_{\text{cat}}^{-1} \text{ h}^{-1}$), which achieves a milestone value for the photocatalytic NO_3^- RR routes and exhibits significant advantage in the other ammonia synthesis routes under ambient condition. The practical application of NO_3^- RR route in simulated wastewater is also developed, which exhibits great expectations for its real industrial application.

●**Page 11 in MS:** By comparing the eight-electron NO_3^- RR and two-electron water splitting reaction, the selectivity for NH_4^+ photosynthesis is determined to be as high as 97.67%. The NH_4^+ synthesis rate and selectivity of the NO_3^- RR on BaO_{NCs} -TNS are superior among the routes under ambient condition, exceeding that of the other photocatalytic NO_3^- RR works and even some leading electrocatalytic NO_3^- RR work (Fig. 4b and Supplementary Table 2). Not surprisingly, the results of this work also exhibit advances in comparison with those of N_2 RR routes at ambient condition, including electrocatalytic, photocatalytic and photoelectrochemical methods.

Fig. 4 Selectivity and long-term stability tests. **a**, NH_4^+ selectivity test from NO_3^- RR versus the other potential products. The related stand curves were provided in Supplementary Figs. 37-39. **b**, Comparison of NH_4^+ production rate and selectivity with different ammonia synthesis routes under ambient conditions. **c**, Calculated activation energy for NO_3^- RR for NH_4^+ synthesis and water splitting for H_2 generation. **d**, Long-term stability of BaONCS-TNS and comparison of total NH_4^+ yield with different NH_4^+ synthesis routes.

Comment 5: The authors wrote “The catalyst exhibits an ultrahigh ammonia photosynthesis rate of $11.97 \text{ mol g}_{\text{metal}}^{-1} \text{ h}^{-1}$ ($89.79 \text{ mmol g}_{\text{cat}}^{-1} \text{ h}^{-1}$) with nearly 100% selectivity.” How is there more g_{cat} than g_{metal} by almost 3 orders of magnitude? Is Ti of the TiO_2 not included as part of g_{metal} ?

Response: Thanks for the comments. Since the active sites were constructed by BaONCS, only the Ba metal mass is calculated in the rate of $\text{g}_{\text{metal}}^{-1}$. To be specific, 0.5 mg of BaONCS-TNS was used for ammonia synthesis efficiency test (Refer the *Experimental section*, Page 21 in MS), which contributed to the rate of $89.79 \text{ mmol g}_{\text{cat}}^{-1} \text{ h}^{-1}$. Then, as shown in the XRF results (Supplementary Fig. 10), 0.75 wt.% of Ba

element is detected in the BaO_{NCS}-TNS composite. Hence, the rate (per Ba element) was calculated to be $11.97 \text{ mol g}_{\text{metal}}^{-1} \text{ h}^{-1} = 89.79 \text{ mmol g}_{\text{cat}}^{-1} \text{ h}^{-1} / 0.75\%$. The formula here is generally applied in calculating the reaction rates, such as the Reference (*J. Am. Chem. Soc.*, **2021**, *143*, 5727-5736). The related description is appended in the revised MS.

●**Page 10 in MS:** Then, as shown in the XRF results (Supplementary Fig. 10), 0.75 wt.% of Ba element is detected in the BaO_{NCS}-TNS composite. Hence, the rate (per Ba element) is calculated to be $11.97 \text{ mol g}_{\text{Ba}}^{-1} \text{ h}^{-1}$.

Comment 6: Page 4: “Key challenge lines in revealing the interfacial structure of active sites for facilitating eight-electron transfer.” I believe “lines” should be “lies”.

Response: Thanks for the suggestion. The typo is revised.

●**Page 4 in MS:** Key challenge **lies** in revealing the interfacial structure of active sites for facilitating eight-electron transfer to increase the selectivity and suppressing the occurrence of side reactions (hydrogen evolution reaction and NO₃⁻ to N₂) to enhance the efficiency.

Comment 7: Figure 1c/d. The inset count labels are not readable unless zoom in to 170%. Perhaps there is a way to make the size distribution image and labels slightly larger.

Response: Thanks for the helpful suggestion. The corresponding images and labels are revised.

●**Page 5 in MS:**

Fig. 1 Structure identification of subnanometric BaO_{NCs} *operando* construction on TNS support. **a-d**, *Quasi-in-situ* HAADF-STEM images showing the evolution course from isolated Ba₂O₃ to subnanometric BaO_{NCs} at the irradiation time of 5 min (**a**), 10 min (**b**), 60 min (**c**) and 120 min (**d**) respectively. The related size distribution is labelled as insets (**c** and **d**), in which the range of both x (0.4-1.0) and y (0-7) axes are set consistently. **e**, HAADF-STEM image (left) and respective elemental mapping images (right) verifying the component of Ba elements on the BaO_{NCs}-TNS surface. **f**, Variation of Ba²⁺ concentration during the *operando* construction of BaO_{NCs} detected by ion chromatography.

Comment 8: “The deconvolution of the Ba 3d XPS spectrum illustrates a perfect fitting to four characteristic peaks at binding energies of 794.61, 792.84, 778.99 and 777.15 eV.” Is the fitting really “perfect”? Perhaps vary the word choice.

Response: Thanks for the suggestion. The corresponding description is revised.

•**Page 6 in MS:** The deconvolution of the Ba 3d XPS spectrum illustrates that the four characteristic peaks were fitted at the binding energies of 794.61, 792.84, 778.99 and 777.15 eV.

Comment 9: “Supplementary Figure 8. Variation of IC signals for Mg₂⁺ (a), Ca₂⁺ (b) and Sr₂⁺ (c) ions during the *operando* construction of corresponding MO_{NCs} on TNTs.” “On” should not be subscripted. Do the authors mean “TNSs” instead of “TNTs”?

Response: Thanks for the helpful suggestion. The corresponding format error and

typo is revised.

●Page 7 in SI:

Supplementary Figure 8. Variation of IC signals for Mg²⁺ (a), Ca²⁺ (b) and Sr²⁺ (c) ions during the operando construction of corresponding MO_{NCS} on TNS.

We really appreciate the comments, the efforts and time the reviewers and editor spent in dealing with our paper. According to your kind suggestions, we try our best to carefully check and promptly revise the manuscript. Hopefully, we have addressed your concerns.

I am looking forward to hearing from you with good news.

Truly yours,

Dr. Fan Dong

University of Electronic Science and Technology of China

REVIEWER COMMENTS

Reviewer #2 (Remarks to the Author):

I again reviewed the manuscript. The author's assumptions are supported by their experimental results and the manuscript is readable. However, I am wondering about the importance of this paper. The reasons are as follows:

The manuscript describes that the BaOx clusters loaded on TiO2 promote photocatalytic NO₃⁻ to NH₄⁺ reduction in water with a sacrificial electron donor (ethylene glycol) under UV irradiation. The authors claimed the almost 100% selectivity for NO₃⁻-to-NH₄⁺ transformation. However, almost 100% selectivity can be achieved on bare TiO2 containing surface oxygen vacancies (Hirakawa, H. et al. Selective nitrate-to-ammonia transformation on surface defects of titanium dioxide photocatalysts. ACS Catal. 2017, 7, 3713), where the oxygen vacancies behave as the NO₃⁻ reduction sites. In addition, Cu particles loaded on TiO2 also promote almost 100% NO₃⁻-to-NH₄⁺ reduction (Kominami, H. et al. Effective photocatalytic reduction of nitrate to ammonia in an aqueous suspension of metal-loaded titanium(IV) oxide particles in the presence of oxalic acid, Catal. Lett. 2001, 76, 31), where Cu particles behave as the reduction sites. Furthermore, Cu-Pd particles loaded on TiO2 also promote selective NO₃⁻-to-NH₄⁺ reduction (Yamauchi, M. et al. JACS 2011, 133, 1150). Given this situation, the present work say "BaOx nanoclusters also behave as the sites for NO₃⁻-to-NH₄⁺ reduction as is the case for the above examples". Sacrificial electron donor is still necessary to promote photocatalytic cycles and to suppress the subsequent oxidation of the formed NH₄⁺. Based on the above, I do not think the results presented here sound to the readers in photocatalysis. I think the significance of this manuscript is no worth publishing in Nat. Commun.

Reviewer #3 (Remarks to the Author):

The authors have addressed all my comments and made extensive revisions, including performing additional experiments and calculations. Below I just provide one minor remark.

1) Figure S36 caption has written "b). The calculations were considered as relaxation when the single imaginary frequency (f/i) was located."

 It is unclear to me what "considered as relaxation" means in this context. Do you mean the transition state was identified when a single imaginary frequency was located and all forces on atoms were zero? Usually "relaxed" geometry corresponds to a local minimum, not a first-order saddle point.

Reviewer #4 (Remarks to the Author):

This work is well-organized with superior originality and of significance to the related research fields in catalysis, energy, materials and environment. Sufficient data analysis was conducted with high quality and the conclusions were well-supported by data. I suggest that this work is qualified to be published.

According to those comments from Reviewer 1, the authors have made solid efforts to append necessary experiments, which further promoted the novelty and reliability of this article. I think that all the comments from Reviewer 1 have been fully addressed and the quality of the newly revised version has been improved. To further help the authors to optimize the manuscript for its publication. The following minor suggestions are provided.

(1) It will be helpful to add some explanation and description on the reasons that the TiO₂ was applied as the catalyst substrate.

(2) In the response to Comment 2 from Reviewer 1, the authors provide some reasons to support the significance of the nitrate reduction route, which are convincing and useful. I suggest integrating these issues into the main text. It should be noted that the corresponding sentences were not completely consistent in the current main text with that in the response letter.

(3) Since the consistent conclusion was achieved by using respective PBE and PBE+U functionals. It may be appropriate to remove the PBE results in Fig. 2e, which guarantees a concise and clear presentation.

(4) In the Abstract, the author said that their photocatalytic performance exceeds some of the electrocatalytic nitrate reduction work. I suggest removing this description. The comparison between this work and the other photocatalytic NO₃RR work is enough to support the advantage of this work.

(5) The investigation of NO₃RR in real wastewater is very indispensable and practical. The authors should list some real conditions where the mixture of nitrate and the other contaminants are emitted, which will be a support to demonstrate the practical value of this nitrate to the ammonia route.

(6) The authors should carefully check each sentence to avoid language mistakes. I can even find some grammar mistakes in the abstract.

Response to Referee 2:

General Comment: I again reviewed the manuscript. The author's assumptions are supported by their experimental results and the manuscript is readable. However, I am wondering about the importance of this paper. The reasons are as follows:

The manuscript describes that the BaOx clusters loaded on TiO₂ promote photocatalytic NO₃⁻ to NH₄⁺ reduction in water with a sacrificial electron donor (ethylene glycol) under UV irradiation. The authors claimed the almost 100% selectivity for NO₃⁻ to NH₄⁺ transformation. However, almost 100% selectivity can be achieved on bare TiO₂ containing surface oxygen vacancies (Hirakawa, H. et al. Selective nitrate-to-ammonia transformation on surface defects of titanium dioxide photocatalysts. ACS Catal. 2017, 7, 3713), where the oxygen vacancies behave as the NO₃⁻ reduction sites. In addition, Cu particles loaded on TiO₂ also promote almost 100% NO₃⁻ to NH₄⁺ reduction (Kominami, H. et al. Effective photocatalytic reduction of nitrate to ammonia in an aqueous suspension of metal-loaded titanium(IV) oxide particles in the presence of oxalic acid, Catal. Lett. 2001, 76, 31), where Cu particles behave as the reduction sites. Furthermore, Cu-Pd particles loaded on TiO₂ also promote selective NO₃⁻ to NH₄⁺ reduction (Yamauchi, M. et al. JACS 2011, 133, 1150). Given this situation, the present work say "BaOx nanoclusters also behave as the sites for NO₃⁻ to NH₄⁺ reduction as is the case for the above examples". Sacrificial electron donor is still necessary to promote photocatalytic cycles and to suppress the subsequent oxidation of the formed NH₄⁺. Based on the above, I do not think the results presented here sound to the readers in photocatalysis. I think the significance of this manuscript is no worth publishing in Nat. Commun.

Response: Thanks for the comment. As Reviewer 2 suggested, the conclusions are as supported by our experimental results, and the manuscript is well organized with readability. We are aware that the main concern from Reviewer 2 is the significance of this work, which is again proposed in this review round. Especially, Reviewer 2 listed many references to indicate that the selective nitrate to ammonia reduction has

been already achieved, which might lower the significance of this work. To address this concern, we must again emphasize that the ammonia synthesis efficiency should be comprehensively evaluated. Rather than **the only indicator of ammonia selectivity, which is not valid for evaluation of the overall ammonia synthesis**, the performance targets of **synthesis rate, selectivity, long-term stability and total yields** should be considered simultaneously, in which our manuscript all exhibits advances in the related ammonia synthesis routes compared to previous reports (Fig. 4b, 4d and Supplementary Table 1-2). We believe that the comprehensive evaluation of ammonia synthesis efficiency in this work is very important and beneficial to promote the sustainable development of the ammonia synthesis area, which we believe could meet the publication standard of *Nature Communications*.

Fig. 4 Selectivity and long-term stability tests. b, Comparison of NH₄⁺ production rate and selectivity with different ammonia synthesis routes under ambient conditions. **d,** Long-term stability of BaO_{NCS}-TNS and comparison of total NH₄⁺ yield with different NH₄⁺ synthesis routes.

Furthermore, the designed subnanometric BaO clusters catalyst for ammonia photosynthesis, which has not been reported yet, can deepen the understanding of the active sites in general catalytic reactions. An essential contribution is made to investigate the dynamic evolution of metal ions in the reaction mixture. The pattern of operando BaO_{NCS} production also promotes the development of fundamental physical chemistry.

As for the sacrificial electron donor issue, it is documented in the revised Manuscript that the “**sacrificial-electron-donor-free**” route has been established by combining nitrate reduction and contaminants’ degradation (formaldehyde, phenol and benzyl alcohol) in simulated wastewater (Fig. 6). **By providing the contaminant**

and nitrate as the only feedstock, this work represents a first route to update the conventional sacrificial electron donor-induced photocatalysis into a new paradigm for both reducing energy consumption and mitigating environmental anxieties.

Page 18: Most importantly, it should be noted that abundant organic contaminants are distributed in many situations of NO_3^- -containing wastewater, which can be utilized as what is called the hole sacrificial agent. Based on this consideration, phenol, benzyl alcohol and formaldehyde were added into the catal. sys. as potential contaminants respectively (Fig. 6b and Supplementary Fig. 50a-50f). It is found that the ammonia synthesis rates and selectivity are all retained, which indicates that the NO_3^- RR route is established in the simulated organic wastewater.

Interestingly, the ammonia synthesis rates are increased in the order of conventional catal. sys. ($3.78 \text{ mmol g}^{-1} \text{ h}^{-1}$) < benzyl alcohol ($5.37 \text{ mmol g}^{-1} \text{ h}^{-1}$) < phenol ($7.64 \text{ mmol g}^{-1} \text{ h}^{-1}$) < formaldehyde ($8.99 \text{ mmol g}^{-1} \text{ h}^{-1}$). It is thus deduced that the hole consumption capacity of these organic contaminants is higher than that of EG, which leaves more electrons to catalyze the NO_3^- RR. Then, the corresponding test of formaldehyde was conducted without EG as the hole sacrificial agent (Fig. 6b and Supplementary Fig. 50g-50h). It is observed that the ammonia synthesis rate is further elevated to $11.14 \text{ mmol g}^{-1} \text{ h}^{-1}$ with 97.82% of selectivity noted, which reveals that the formaldehyde can act as the “hole sacrificial agent” more efficiently than that of EG. Based on the organic contaminant investigation in the simulated wastewater, it is clarified that the NO_3^- RR route can be developed as a “sacrificial-agent-free” technology for the application of ammonia synthesis in wastewater coupled with the organic pollutants’ oxidation, which demonstrates scientific significance in both areas of environmental remediation and energy conversion.

Fig. 6b, Ammonia synthesis rates and selectivity evaluation by adding different types of simulated wastewater into the catalysis system (Catal. Sys.), including the organic matter (phenol, benzyl alcohol and formaldehyde), cation (Co^{2+} , Ni^{2+} and Cd^{2+}) and anion (SO_4^{2-} , PO_4^{3-} and CO_3^{2-}) contaminants correspondingly. As for the condition of formaldehyde, the catalytic tests were conducted with and without EG respectively.

Response to Referee 3:

General Comment: The authors have addressed all my comments and made extensive revisions, including performing additional experiments and calculations.

Below I just provide one minor remark.

Response: Thanks for the helpful comments and suggestions.

Comment 1: Figure S36 caption has written "b). The calculations were considered as relaxation when the single imaginary frequency (f/i) was located."

 It is unclear to me what "considered as relaxation" means in this context. Do you mean the transition state was identified when a single imaginary frequency was located and all forces on atoms were zero? Usually "relaxed" geometry corresponds to a local minimum, not a first-order saddle point.

Response: Thanks for the suggestion. This confusing description is now revised in the Supplementary Information.

Supplementary Figure 36. Calculated reaction coordinates of NO_3^- reduction (**a**) and water splitting (**b**). The transition state was identified when a single imaginary frequency was located and all forces on atoms were zero. The blue, red, green, light grey and pink spheres depict Ti, O, Ba, N and H atoms, respectively.

Response to Referee 4:

General Comment: This work is well-organized with superior originality and of significance to the related research fields in catalysis, energy, materials and environment. Sufficient data analysis was conducted with high quality and the conclusions were well-supported by data. I suggest that this work is qualified to be published.

According to those comments from Reviewer 1, the authors have made solid efforts to append necessary experiments, which further promoted the novelty and reliability of this article. I think that all the comments from Reviewer 1 have been fully addressed and the quality of the newly revised version has been improved. To further help the authors to optimize the manuscript for its publication. The following minor suggestions are provided.

Response: Thanks for the nice comments and helpful suggestions.

Comment 1: It will be helpful to add some explanation and description on the reasons that the TiO₂ was applied as the catalyst substrate.

Response: Thanks for the comments. The respective description has been added in the modified Manuscript.

Page 4: Here, we demonstrate a general strategy to accomplish the *operando* construction of subnanometric alkaline-earth oxide clusters (MO_{NCS}, M=Mg, Ca, Sr or Ba) as the active sites due to the nontoxicity and low price of alkaline-earth metals. Also, the widely investigated TiO₂ nanosheets (TNS) is applied as the substrate since it is simple to be fabricated and characterized.

Comment 2: In the response to Comment 2 from Reviewer 1, the authors provide some reasons to support the significance of the nitrate reduction route, which are convincing and useful. I suggest integrating these issues into the main text. It should be noted that the corresponding sentences were not completely consistent in the current main text with that in the response letter.

Response: Thanks for the comments. According to the suggestions of Reviewer 4 and

the editorial team, the importance for nitrate to ammonia reduction is discussed in the revised Manuscript.

Page 3: Another virtue of using NO_3^- feedstock lies in its rich distribution in wastewater. The abundant nitrate in wastewater offers sufficient reactants for NO_3^- reduction reaction (NO_3^- RR) routes. Instead of the partial reduction of NO_3^- to N_2 for its purification, the eight-electron transfer reaction for NO_3^- to NH_4^+ synthesis provides a new opportunity for the value-added conversion of contaminative NO_3^- into ammonia as an economically competitive product.

Comment 3: *Since the consistent conclusion was achieved by using respective PBE and PBE+U functionals. It may be appropriate to remove the PBE results in Fig. 2e, which guarantees a concise and clear presentation.*

Response: Thanks for the suggestion. The PBE result in Fig, 2e has been removed in the revised Manuscript.

Fig. 2e Calculated planar average potential energy profile using PBE+U correction. Inset: calculated charge difference distribution at the $\text{BaO}_{\text{NCS}}/\text{TNS}$ interface, in which charge accumulation is marked in blue and charge depletion is marked in yellow. The isosurface was set to $0.005 \text{ eV } \text{\AA}^{-3}$.

Comment 4: *In the Abstract, the author said that their photocatalytic performance exceeds some of the electrocatalytic nitrate reduction work. I suggest removing this description. The comparison between this work and the other photocatalytic NO_3^- RR work is enough to support the advantage of this work.*

Response: Thanks for the comments. This respective description is removed in the revised Manuscript.

Page 2: A total ammonia yield of 0.78 mmol within 72 hours is achieved, which exhibits significant advantage in the area of photocatalytic NO_3^- RR.

Comment 5: The investigation of NO_3^- RR in real wastewater is very indispensable and practical. The authors should list some real conditions where the mixture of nitrate and the other contaminants are emitted, which will be a support to demonstrate the practical value of this nitrate to the ammonia route.

Response: Thanks for the comments. Some emission conditions where the mixture of nitrate and the other contaminants are emitted are appended in the revised Manuscript.

Page 17: Most importantly, it should be noted that abundant organic contaminants are distributed in many NO_3^- -containing emission conditions such as the agricultural and chemical wastewater degradation and drinking water purification, in which the organic contaminants can be utilized as what is called the hole sacrificial agent.

Comment 6: The authors should carefully check each sentence to avoid language mistakes. I can even find some grammar mistakes in the abstract.

Response: Thanks for the comments. This whole manuscript has been checked to avoid language mistakes. Here are some examples:

Page 2: The limitation of inert N_2 molecules with its high dissociation energy has ignited new research interests in probing other nitrogen-containing species for ammonia synthesis.

Page 2: The investigation of the molecular-level reaction mechanism reveals that the unique active interface between the subnanometric clusters and TiO_2 substrate is beneficial for the nitrate activation and dissociation.

Page 2: The practical application of NO_3^- RR route in simulated wastewater is also developed, which demonstrates great potential for its industrial application.

Page 3: Inspired by the natural microbial N_2 fixation, artificial electro-/photo-/photoelectrochemical nitrogen reduction reactions (N_2 RRs) for NH_3 synthesis as an alternative, have attracted tremendous research interest.

Page 6: Then, equilibrium is reached to guarantee the subnanometric size of the

BaO_{NCS}, thereby preventing excessive agglomeration and further growth.

Page 11: K¹⁴NO₃ and K¹⁵NO₃ solutions are used as N sources.

We really appreciate the comments, the efforts and time the reviewers and editor spent in dealing with our paper. According to your kind suggestions, we try our best to carefully check and promptly revise the manuscript. Hopefully, we have addressed your concerns.

I am looking forward to hearing from you with good news.

Truly yours,

Dr. Fan Dong

University of Electronic Science and Technology of China